# CD74 promotes the formation of an immunosuppressive tumor microenvironment in triple-negative breast cancer in mice by inducing the expansion of tolerogenic dendritic cells and regulatory B cells

**Bianca Pellegrino[1], Keren David[1], Stav Rabani[1], Bar Lampert[1], Thuy Tran[2], Edward Doherty[2], Marta Piecychna[2], Roberto Meza-Romero[3,4,5], Lin Leng[2], Dov Hershkovitz[6], Arthur A. Vandenbark[3,4,5], Richard Bucala[2], Shirly Becker-Herman[1], Idit Shachar** [1] *

1 Department of Systems Immunology, Weizmann Institute of Science, Rehovot, Israel, 2 Yale Cancer Center and School of Medicine, New Haven, Connecticut, United States of America, 3 Neuroimmunology Research, VA Portland Health Care System, Portland, Oregon, United States of America, 4 Department of Neurology, Oregon Health & Science University, Portland, Oregon, United States of America, 5 Department of Molecular Microbiology and Immunology, Oregon Health & Science University, Portland, Oregon, United States of America, 6 Insitute of Pathology, Sourasky Medical Center, Tel Aviv, Israel

* idit.shachar@weizmann.ac.il

## Abstract

CD74 is a cell-surface receptor for the cytokine macrophage migration inhibitory factor (MIF). MIF binding to CD74 induces a signaling cascade resulting in the release of its cytosolic intracellular domain (CD74-ICD), which regulates transcription in naïve B and chronic lymphocytic leukemia (CLL) cells. In the current study, we investigated the role of CD74 in the regulation of the immunosuppressive tumor microenvironment (TME) in triple-negative breast cancer (TNBC). TNBC is the most aggressive breast cancer subtype and is characterized by massive infiltration of immune cells to the tumor microenvironment, making this tumor a good candidate for immunotherapy. The tumor and immune cells in TNBC express high levels of CD74; however, the function of this receptor in the tumor environment has not been extensively characterized. Regulatory B cells (Bregs) and tolerogenic dendritic cells (tol-DCs) were previously shown to attenuate the antitumor immune response in TNBC. Here, we demonstrate that CD74 enhances tumor growth by inducing the expansion of tumor-infiltrating tol-DCs and Bregs. Utilizing CD74-KO mice, Cre-flox mice lacking CD74 in CD23+ mature B cells, mice lacking CD74 in the CD11c+ population, and a CD74 inhibitor (DRQ), we elucidate the mechanism by which CD74 inhibits antitumor immunity. MIF secreted from the tumor cells activates CD74 expressed on DCs. This activation induces the binding of CD74-ICD to the SP1 promotor, resulting in the up-regulation of SP1 expression. SP1 binds the IL-1β promotor, leading to the down-regulation of its transcription. The reduced levels of IL-1β lead to decreased antitumor activity by allowing expansion of the tol-

**Data Availability Statement:** All relevant data are within the paper and its Supporting Information files. FCS files uploaded to FlowRepository (http://flowrepository.org/id/FR-FCM-Z8ES).

**Funding:** This research was supported by Israel Science Foundation grant no. 1642/20; the Department of Veterans Affairs, Veterans Health Administration, Office of Research and Development, Biomedical Laboratory Research and Development Merit Review Award 2I01 BX000226 (to AAV), BLR&D Merit Review for Pre-IND studies of Drugs and Biologics Award 5I01 BX005112 (to AAV) and Senior Research Career Scientist Award 1IK6BX004209 (to AAV). The contents do not represent the views of the Department of Veterans Affairs and the US Government NIH grants 1R01-AR-078334, 5R01- AI110452, and the ACR RRF (to RB) The funders had no role in study design, data collection and analysis, decision to publish, or preparation of the manuscript.

**Competing interests:** AAV, RM-R and OHSU have a significant financial interest in Artielle ImmunoTherapeutics, Inc., a company that may have a commercial interest in the results of this research and technology. This potential conflict of interest has been reviewed and managed by the OHSU and VA Portland Health Care System Conflict of Interest in Research Committees.

**Abbreviations:** APC, antigen-presenting cell; BC, breast cancer; ChIP-qPCR, chromatin immunoprecipitation qPCR; CLL, chronic lymphocytic leukemia; CPD, Cell Proliferation Dye; DC, dendritic cell; DSG, disuccinimidyl glutarate; ER, estrogen receptor; HER2, human epidermal growth factor receptor 2; ME, microenvironment; MIF, migration inhibitory factor; MIT, mithramycin; mrMIF, mouse recombinant MIF; PR, progesterone receptor; TAA, tumor-associated antigen; TIL, tumor-infiltrating lymphocyte; TME, tumor microenvironment; TNBC, triple-negative breast cancer; tol-DC, tolerogenic dendritic cell; WT, wild-type.

DC, which induces the expansion of the Breg population, supporting the cross-talk between these 2 populations. Taken together, these results suggest that CD74+ CD11c+ DCs are the dominant cell type involved in the regulation of TNBC progression. These findings indicate that CD74 might serve as a novel therapeutic target in TNBC.

## Introduction

Breast cancer (BC) is the most frequent malignancy affecting women worldwide [1]. BC is a heterogeneous disease comprised of different subtypes. The classification of BC depends on the expression of 3 biomarkers: estrogen receptors (ERs), progesterone receptors (PRs), and human epidermal growth factor receptor 2 (HER2). BCs that are negative for ER, PR, and HER2 are known as triple-negative breast cancer (TNBC) [2]. TNBC is regarded as the most aggressive BC subtype, and its clinical features include high invasiveness, high metastatic potential, the propensity to relapse, and poor prognosis [3,4]. TNBC has been shown to feature a unique microenvironment (ME). TNBC cells manipulate their ME to become immunosuppressive, making this tumor a good candidate for immunotherapy [5].

Dendritic cells (DCs) are antigen-presenting cells (APCs) essential for the regulation of innate and adaptive immune responses. The multiple functions of DCs in immune regulation mirror their complexity and their heterogeneous subsets with different lineages, locations, phenotypes, and functional plasticity [6]. DCs play an important role in tumor immunity and are able to cross-present tumor-associated antigens (TAAs) to T cells. By manipulating their ME, tumor cells can perturb DC functions, reduce T cell activation, and, potentially, the induction of T cell tolerance to TAAs [7]. The tumor can also empower immune-regulatory transcriptional programs that limit the DC-mediated production of pro-inflammatory cytokines and increase the release of IL-10 and indoleamine dioxygenase-1 (IDO1), which facilitate immunosuppression. DCs that produce IL-10 enforce T cell anergy and are termed tolerogenic dendritic cells (tol-DCs). IL-10 expression in DCs is considered a tolerogenic signature resulting in the induction of Tregs [8].

B cells also play an important function in the tumor microenvironment (TME). In addition to their role in the regulation of the humoral immune responses and their ability to produce antibodies and cytokines, some B cell populations, known as regulatory B cells (Bregs), have regulatory properties that are crucial for the maintenance of immune tolerance [9]. Bregs exert their function predominantly via the release of IL-10. The immune suppressive function of Bregs involves multiple mechanisms, including skewing T cell differentiation, induction, and maintenance of Tregs, as well as suppression of pro-inflammatory cells [10].

CD74 (Ii chain) is a non-polymorphic type II transmembrane protein expressed mainly on the surface of APCs and was initially thought to function solely as an MHC class II chaperone [11]. A small portion of the CD74 molecules undergo posttranslational modifications that enable their cell surface expression [12]. Cell surface CD74 serves as a receptor for ligands of the macrophage migration inhibitory factor (MIF) family, which includes the cytokines MIF-1 (MIF) and MIF-2/D-dopachrome tautomerase (DDT) [13]. Upon MIF binding, CD74 forms a cell surface complex with CD44, which takes part in the MIF-induced signaling cascade. The signaling pathway involves Syk tyrosine kinase and PI3K/Akt activation, leading to CD74 intramembrane cleavage and the release of the CD74 intracellular domain (CD74-ICD). CD74-ICD translocates to the nucleus, where it induces cell proliferation and survival of B cells [14–20]. Interestingly, both MIF and CD74 have been associated with tumor progression

and metastasis. It was reported that MIF mRNA is overexpressed in various tumors [21,22] and that MIF has also been associated with the growth of malignant cells [23]. Many studies have demonstrated that CD74 expression is regulated in various cancers [24–29], including chronic lymphocytic leukemia (CLL) [30,31], and correlates with poor prognosis. In particular, CD74 expression is up-regulated in patients with TNBC compared to other BC subtypes and is associated with lymph node metastasis, leading to a worsening of overall survival [32,33]. CD74 expression has also been suggested to serve as a prognostic factor in many of these cancers, with higher relative expression of CD74 behaving as a marker of tumor progression [34]. Although it is well known that CD74 plays a crucial role in hematological malignancies such as CLL [35] and that its expression correlates with a poor prognosis, its function and mechanism in TNBC are incompletely understood.

In the current study, we followed the role of CD74 in the regulation of the TME in TNBC. We demonstrate that CD74 expressed on CD11C$^+$ cells regulate tumor growth through the control of the cross-talk between the tumor-infiltrating tol-DCs and Bregs. These findings suggest CD74 as a novel therapeutic target in TNBC.

## Results

### CD74 regulates tumor load through the control of immune-suppressive populations in the TNBC murine model

To determine whether CD74 plays a role in the functionality of the tumor-infiltrating immune cells, we first analyzed its expression on these cells. Biopsies from the breasts of healthy individuals and TNBC patients were analyzed for the presence of B cells and DCs and CD74 expression. As shown in Fig 1A, massive infiltration of B cells and DCs was detected in the TME compared to the tissue of healthy donors. These cells highly expressed CD74 in TNBC tissues (Fig 1A), suggesting a role for CD74 in the TNBC ME.

Next, CD74 expression was analyzed in a murine model of TNBC. E0771 murine TNBC cells were orthotopically injected into C57BL/6, and CD74 expression on immune cells infiltrating the tumor was determined by FACS analysis. CD74 was widely expressed on immune cells. Furthermore, its expression was significantly higher on the tumor-infiltrating B cells and DCs compared to its levels on the peripheral splenic populations (Fig 1B–1D), suggesting a role for CD74 expressed on these cells in the TNBC ME.

To determine the in vivo role of CD74 in the ME of TNBC, E0771 murine TNBC cells were orthotopically injected into C57BL/6 or CD74 deficient (CD74$^{-/-}$) mice. Tumor size was monitored every 5 days from the day of injection, and mice were killed on day 21. As shown in Fig 2A, the absence of CD74 significantly inhibited tumor development and growth. Furthermore, the difference was found to be significant according to the area under the curve graphing tumor progression (inset in Fig 2A). In addition, tumor size at the end of the experiment was markedly smaller in mice lacking CD74, as revealed in Fig 2B and 2C.

TNBC cells reprogram their ME towards an immunosuppressive phenotype by inducing the secretion of IL-10 in the various immune cell populations [5]. We therefore next analyzed the APCs and T cells in the TME derived from wild-type (WT) and CD74-deficient mice. DCs positively or negatively regulate the antitumor immune response according to the cytokines released and the expression of costimulatory molecules able to bind their T cell counterparts to induce their priming [36]. Thus, DCs in the TME in WT and CD74-deficient mice were analyzed for their numbers and functionality by FACS analysis. As shown in Fig 2D, a significantly higher percentage of CD74-deficient DCs were observed in the TME compared to the TME of WT mice. These accumulated cells expressed higher levels of CD80 (Fig 2E) and a lower percentage of IL-10 (the gating strategy is shown in Figs 2F, S1A–S1I, and S2A and S10 Data),

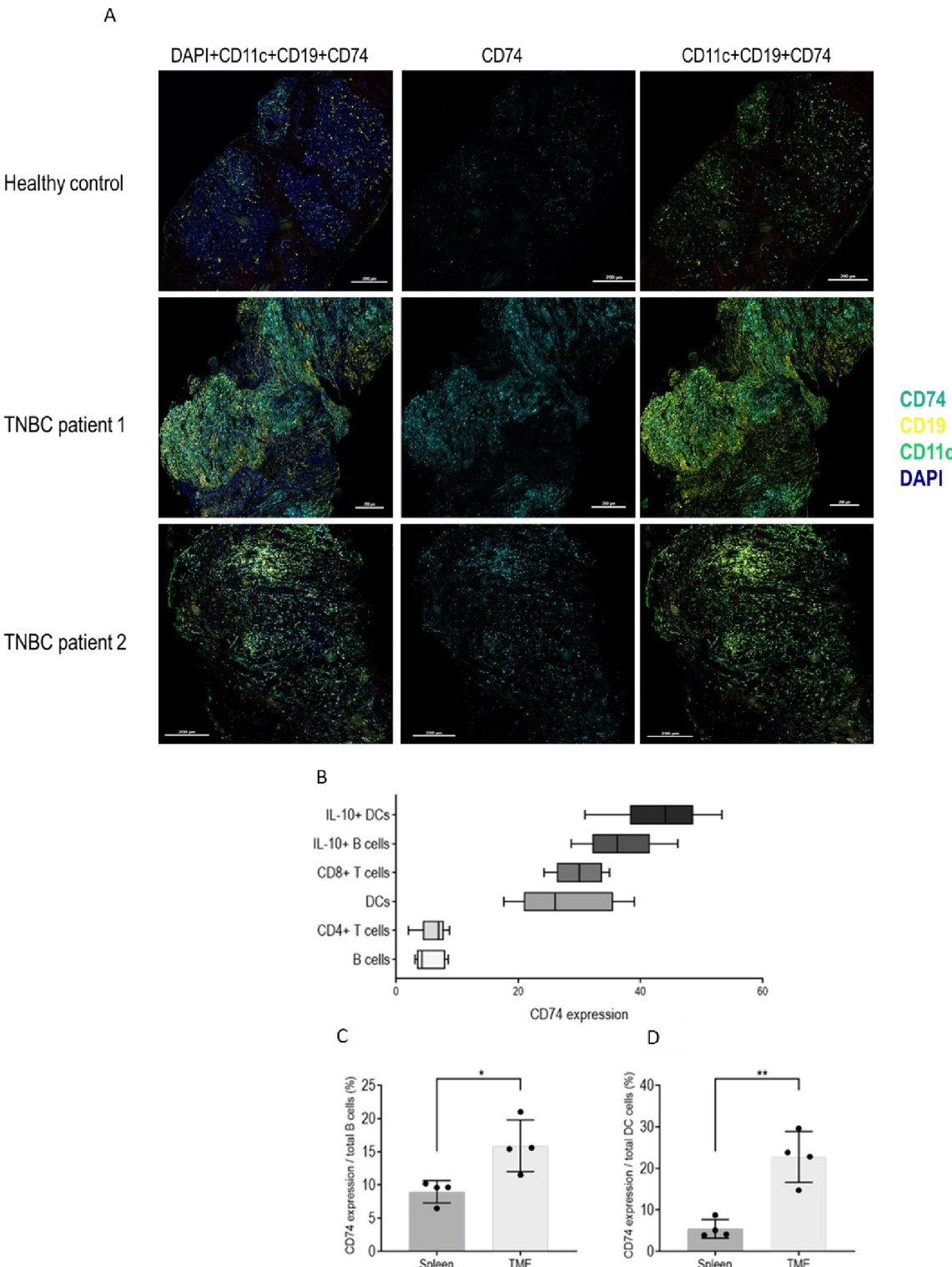

**Fig 1. CD74 expression is up-regulated in the TME. (A)** Representative Opal Multiplex IHC staining for: CD74 (cyano) (middle panels), CD19 (yellow), and CD11c (green) (right panels), in healthy breast tissue (top), and in 2 TNBC patients (middle, bottom). DAPI staining is shown on the left. **(B–D; S1 Data)** Six weeks old C57BL/6 female mice were injected with $5 * 10^5$ E0771 cells into each of the fourth mammary pads (total of 2 mammary pads per mouse). Mice were killed, and total PBMCs from the tumor site and spleen were activated with PIM, and then analyzed by flow cytometry. Dead cells were excluded from analysis by Zombie Live/Dead staining. **(B)** Violin plot analysis depicting the relative expression level of CD74 in several immune cell populations including B cell, CD4$^+$ and

CD8⁺ T cells, DCs, Bregs and tol-DCs. **(C)** Graph showing the mean percentage and SD of CD74 expression on B cells in the spleen and in the TME (B cells in the spleen, *n* = 4; B cells in the TME, *n* = 4). **(D)** Column histogram represents the mean percentage and SD of CD74 expression on DCs in the spleen and in the TME (DCs in the spleen, *n* = 4; DCs in the TME, *n* = 4). * *p* < 0.05, ** *p* < 0.005. Breg, regulatory B cell; DC, dendritic cell; PBMC, peripheral blood mononuclear cell; TME, tumor microenvironment; TNBC, triple-negative breast cancer; tol-DC, tolerogenic dendritic cell.

suggesting that the CD74-deficient DC in the TME were more immunogenic and less tolerogenic. Furthermore, the percentage and frequency of Bregs (gating strategy appears in S1J–S1M Fig; frequency appears in S2B Fig and S10 Data), IL-10⁺ macrophages, regulatory T cells, CD4⁺ T cells, and CD8⁺ exhausted T cells were down-regulated. No differences were detected in the frequency of CD8⁺ T cells or CD8⁺ CD62⁺ T cells, or in the cytotoxic activity of CD8⁺ T cells in the TME of CD74 KO mice (S2C–S2I Fig and S10 Data).

Since deficiency of CD74 results in a reduced number of mature B cells and CD4⁺ T cells [37], we next wished to determine whether the reduced tumor load detected in the CD74-deficient mice results from the lack of immune cells that are able to control the antitumoral

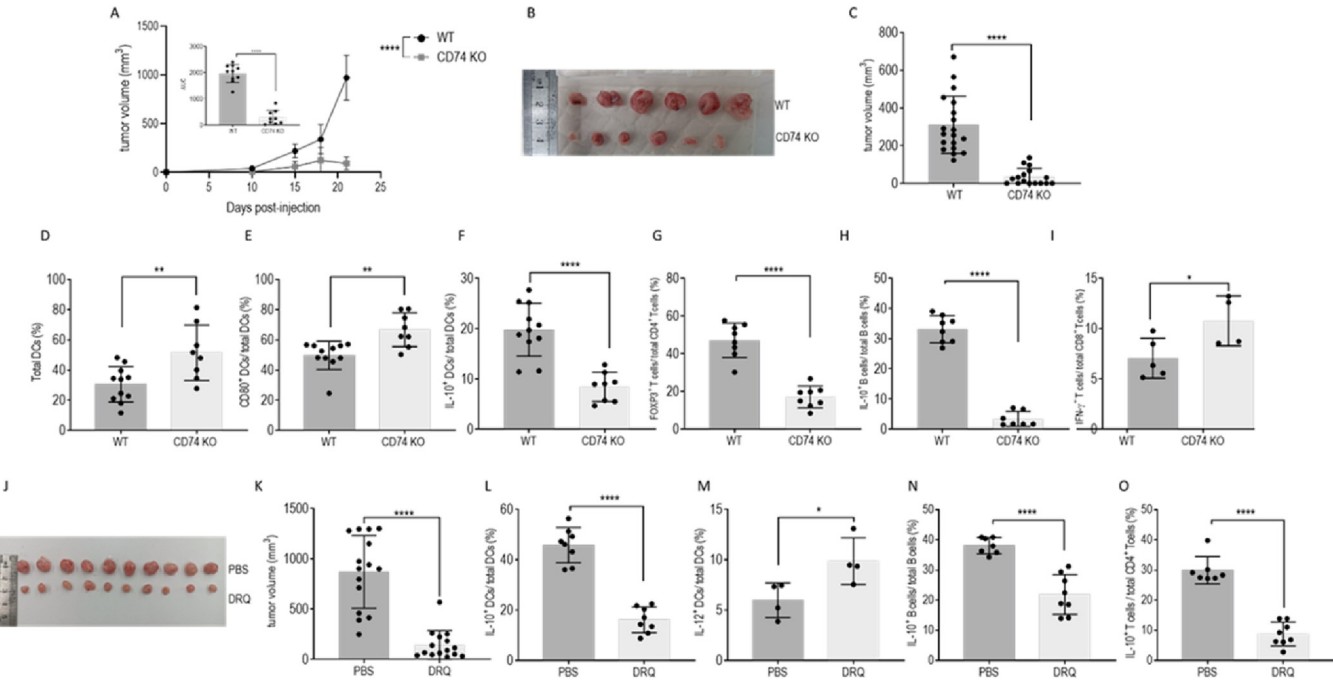

**Fig 2. CD74 regulates tumor progression. (A–I; S2 Data)** Six-week-old C57BL/6 and CD74⁻/⁻ female mice were injected with 5 * 10⁵ E0771 cells into each of the fourth mammary pads (total of 2 mammary pads per mouse). **(A)** Tumor size was measured every 5 days. The difference in tumor growth was found to be significant according to the area under the curve graphing tumor volume (inset in Fig 1A) **(B, C)** After 21 days, mice were killed, and tumors were removed and measured; (*n* = 34), each dot on the graph represents a tumor. **(D–I)** Total PBMCs from the tumor site were activated with PIM (PMA (phorbol 12-myristate 13-acetate), ionomycin, monensin) and then analyzed by flow cytometry. Dead cells were excluded from analysis by Zombie Live/Dead staining. DCs were analyzed for CD45, CD11c, and CD80 expression after excluding LY6-C⁺, F4/80⁺, and CD19⁺ cells. **(D)** Mean percentage and SD of DCs in the tumor site (WT, *n* = 11; CD74⁻/⁻, *n* = 8). **(E)** CD80⁺ DCs analyzed as percentage of total DCs (WT, *n* = 11; CD74⁻/⁻, *n* = 8). **(F)** tol-DCs were analyzed for IL-10 expressing cells as a fraction of total DCs. Graph shows the mean percentage and SD of IL-10⁺ DCs out of total DCs (WT, *n* = 11; CD74KO, *n* = 8). **(G)** Mean percentage and SD of FOXP3⁺ T cells out of total CD4⁺ T cells (WT, *n* = 8; CD74KO, *n* = 8). **(H)** Mean percentage and SD of IL-10⁺ B cells out of total B cells (WT, *n* = 8; CD74, KO *n* = 7). **(I)** Mean percentage and SD of IFN-γ⁺ T cells out of total CD8⁺ T cells (WT, *n* = 5; CD74KO, *n* = 4). **(J-O)** Female 6-week-old C57BL/6 mice were injected with 5 * 10⁵ E0771 cells into each of the fourth mammary pads and injected intravenously with DRQ on days 10, 11, 12, 13, 14, after tumor onset. **(J, K; S2 Data)** After 21 days, tumor sizes were measured and recorded (*n* = 29)**. (L)** IL-10⁺ DCs shown as the percentage of total DCs (for PBS, *n* = 8; DRQ, *n* = 8). **(M)** IL-12⁺ DCs shown as the percentage of total DCs (for PBS, *n* = 4; DRQ, *n* = 4). **(N)** Mean percentage and SD of IL-10⁺ B cells out of total B cells (for PBS, *n* = 7; DRQ, *n* = 8). **(O)** Mean percentage and SD of Tregs out of total CD4⁺ cells (for PBS, *n* = 7; DRQ, *n* = 8). Raw data represented in S2 Data. * *p* < 0.05, ** *p* < 0.005, ***P < 0.0005, ****p < 0.00005. DC, dendritic cell; PBMC, peripheral blood mononuclear cell; tol-DC, tolerogenic dendritic cell; WT, wild type.

response, or whether it is due to the role of CD74 as a MIF receptor, regulating the function of immune cells. To address this question, CD74 function was blocked with DRhQ, a partial MHC class II construct, which inhibits MIF binding to CD74 on the cell surface, though not its role in antigen presentation (to affect antigen presentation, DRhQ would have to bind CD74 in the ER, the site of CD74 and nascent Class II association, and be tightly bound to the chaperone once reaching the endosomal compartment) [38–40]. TNBC cells were injected into the mice, and starting from day 10, mice were intravenously treated for 5 consecutive days (10 to 14), with either DRQ or vehicle (saline) control. Blocking CD74 reduced tumor growth and tumor volume (Fig 2J and 2K). Analysis of immune cells in the TME showed that this treatment elevated the percentage of immunogenic DC, resulting in a down-regulation of tolerogenic IL-10$^+$ DC cells (tol-DCs; Figs 2L and S3A and S11 Data), and up-regulation of the IL-12-expressing DC (Fig 2M) in the TME. Furthermore, a decrease in infiltrating Bregs and Tregs was observed in the DRQ-treated mice (Figs 2N, 2O, and S3B–S3D and S11 Data), as well as an up-regulation in the frequency of CD8$^+$ T cells (S3E Fig and S11 Data). These results suggest that MIF binding to CD74 positively regulates the tumor-suppressive ME in the TNBC.

## MIF induces the CD74 suppression of immune cells of the TME

To determine whether CD74 can induce an in vitro expansion of tol-DC and Breg in the presence of cancer cells, a coculture experiment was performed. Purified splenocytes derived from naïve WT and CD74 KO mice were cultured either alone or in the presence of E0771 cells, as shown in the flow chart in Fig 3A. Cells were analyzed by FACS after 24 hours. The presence of cancer cells induced the expansion of both the Breg (Fig 3B) and the tol-DC population (Fig 3C). This expansion was abrogated in the presence of immune cells lacking CD74, emphasizing the importance of CD74 as an immunosuppressive pro-oncogenic factor.

Since MIF is the ligand of CD74, we next determined whether MIF secreted from TNBC cells contributes to the expansion of immunosuppressive DCs and Bregs. Purified DC or B cells were separately cultured in presence of E0771 cells and activated with mouse recombinant MIF (mrMIF) or vehicle for 24 hours (Fig 3D). As shown in Fig 3E and 3F, MIF induced a modest expansion of IL-10-positive tol-DCs (Fig 3E) and Bregs (Fig 3F). Since cancerous cells endogenously produce and release MIF, we wished to directly confirm that the source of the MIF that regulates immunosuppressive cell expansion, is secretion from the tumor cells. Using a neutralizing MIF antibody or an MIF antagonist would not distinguish between MIF present in the media and secreted by the E0771 cells themselves. Instead, directly reducing MIF expression in the E0771 cells and its presence in the media by adding a low percentage of serum resulted in complete MIF depletion. To this end, MIF expression in the E0771 cells was knocked down by MIF siRNA, then cultured with DC or B cells, and their phenotype was analyzed (Fig 3G and 3H). Down-regulation of MIF expression resulted in significantly reduced expansion of tol-DCs (Fig 3I) and Bregs (Fig 3J). Thus, the MIF expression by the malignant cells plays a crucial role in the regulation of tol-DC and Breg expansion.

## The effect of CD74 on tumor growth is intrinsic to dendritic cells

To determine which APCs (B or DCs) lacking CD74 regulate tumor growth, CD74 was specifically down-regulated in mature B cells or DCs using conditional ablation in CD74$^{-/-}$ Cre-flox mice (cKO). Specifically, WT mice lacking CD74 uniquely in CD23$^+$ mature B cells, and mice lacking CD74 in the CD11c$^+$ DC population were injected with E0771 tumor cells, tumor size was monitored weekly, and mice were killed on day 21. The lack of CD74 in mature B cells did not affect the tumor load (Fig 4A) nor the phenotype of the infiltrating DCs. Similar IL-10

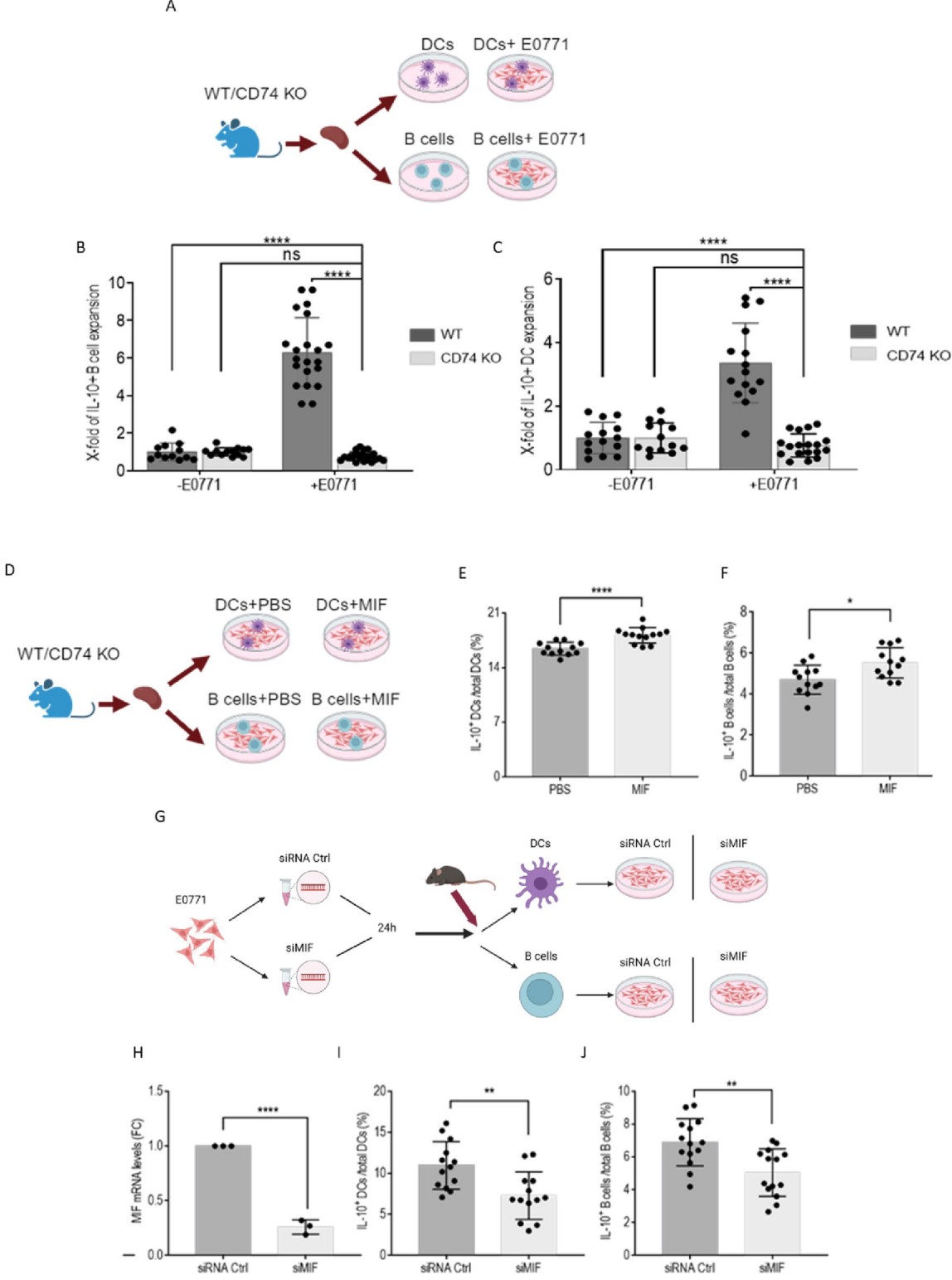

**Fig 3. The MIF-CD74 axis regulates tol-DC and Breg expansion. (A–C; S3 Data)** **(A)** B cells and DCs were isolated from spleens of naïve C57BL/6 and CD74[-/-] mice and cultured either alone or in coculture with E0771 cells at a 1:5 (B cells: E0771) or 1:3 (DCs: E0771) ratio. After 24 hours, B cells or DCs were collected and analyzed for CD19 and IL-10 expression or for CD11c and IL-10 expression, respectively, after excluding LY6-C[+], F4/80[+,] and CD19[+] cells, by flow cytometry. Cells were activated with PIM prior to FACS staining. Dead cells were excluded from analysis by Zombie Live/Dead staining. **(B)** Fold change of IL10[+] DCs expansion out of total DCs, under

the different culture conditions (WT, $n = 15$; CD74KO, $n = 15$), each dot represents a mouse. **(C)** Fold change of Breg expansion out of B cells, under the different culture conditions (WT, $n = 21$; CD74KO, $n = 21$). **(D–F; S3 Data) (D)** Splenic B cells and DCs were purified from IL-10 vert-x mice and cultured with E0771 cells at a 1:5 ratio, in the presence of absence of mrMIF for 24 hours. **(E)** Mean percentage and SD of IL10+ DCs out of total DCs (WT+PBS, $n = 12$; WT+MIF, $n = 13$). **(F)** Mean percentage and SD of Bregs out of B cells (WT+PBS, $n = 12$; WT+MIF, $n = 12$). **(G–J; S3 Data) (G)** E0771 cells were transfected with siRNA MIF or siCtrl. Splenic B cells and DCs were purified from IL-10 vert-x mice and added to the transfected E0771 at a 1:5 or 1:3 ratio for 24 hours. **(H)** MIF mRNA levels were analyzed by qRT-PCR. Graphs present fold change (siRNA/siCtrl) of the chosen gene ($n = 3$). **(I)** Mean percentage and SD of IL10+ DCs out of total DCs (WT+Sico E0771, $n = 13$; WT+siMIF, $n = 13$). **(J)** Bar graphs representing the mean percentage and SD of Bregs out of B cells (WT+Sico E0771, $n = 14$; WT+siMIF, $n = 14$). * $p < 0.05$, ** $p < 0.005$, ***$P < 0.0005$, ****$p < 0.00005$. Breg, regulatory B cell; DC, dendritic cell; MIF, migration inhibitory factor; mrMIF, mouse recombinant MIF; qRT-PCR, quantitative reverse transcription PCR; tol-DC, tolerogenic dendritic cell; WT, wild-type.

(Fig 4B) and IL-12 (Fig 4C) levels were detected in DCs derived from both WT mice and animals deficient in CD74 in their B cell population. However, deficiency of CD74 in the mature B cell population resulted in a significant decrease of Bregs in the TME (Fig 4D), suggesting a direct role for CD74 in the regulation of Bregs. To further investigate the role of CD74 knockdown in mature CD23+ B cells on DCs, in vitro coculture of splenic DC cells derived from WT or CD23 cKO with or without E0771 cells was performed. Deficiency of CD74 in the mature B cell population had no effect on IL-10 release from naïve or cancer-activated DCs (Fig 4E), while the coculture of B cells with E0771 cells led to a weaker induction of Bregs (Fig 4F).

To follow the role of CD74 in DCs, CD74 expression was down-regulated in CD11C+ cells. We first validated the specificity of CD74 deletion to DCs by analyzing the accumulation of the CD11c+ population and their CD74 expression (S4A-S4H Fig and S12 Data) and by

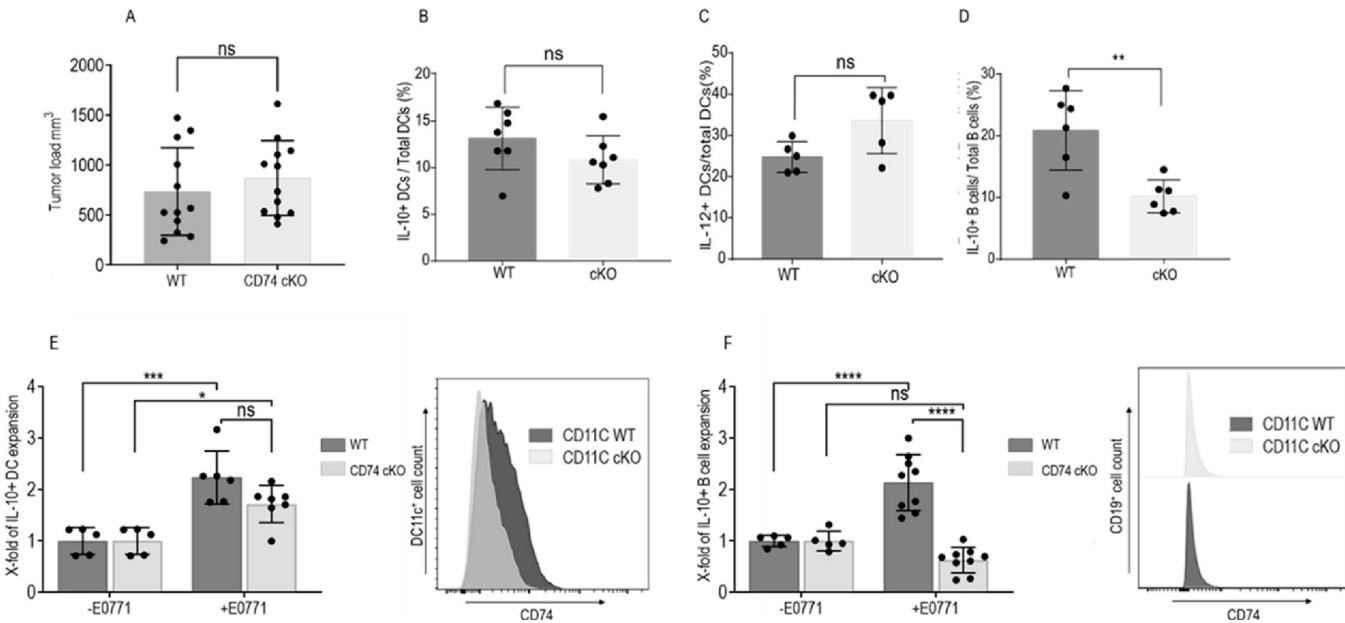

**Fig 4. Deficiency of CD74 in mature B cells does not affect tumor proliferation. (A–D; S4 Data)** Female 6-week-old CD23-Cre x CD74flox x CD74 flox mice were injected with $5 * 10^5$ E0771 cells into each of the fourth mammary pads. **(A)** After 21 days, tumor sizes were measured and recorded ($n = 22$). Mice were killed, and tumors were harvested, processed to a single cell suspension, and total PBMCs from the tumor site were isolated. Cells were then activated with PIM and analyzed by flow cytometry. Dead cells were excluded from analysis by Zombie Live/Dead staining. DCs were analyzed for CD45, CD11c, and IL-10 expression after excluding LY6-C+, F4/80+,' and CD19+ cells. Bregs were analyzed for CD19 and IL-10. **(B)** IL-10+ DCs were analyzed as the percentage of total DCs (for the WT, $n = 8$; conditional cKO, $n = 7$). **(C)** Mean percentage and SD of IL-12+ DCs out of total DCs (for the WT, $n = 5$; cKO, $n = 5$). **(D)** Mean percentage and SD of IL-10+B cells out of total CD19+ cells (for the WT, $n = 6$; cKO, $n = 6$). **(E, F; S4 Data)** Splenic B cells and DCs were cultured with E0771 cells at a 1:5 ratio for 24 hours. **(E)** Mean percentage and SD of IL10+ DCs out of total DCs (WT, $n = 5$; CKO, $n = 5$). **(F)** Mean percentage and SD of Bregs out of B cells (WT, $n = 7$; cKO, $n = 7$). The histograms represent the expression of CD74 in the DC and B cell populations. ns $p > 0.05$, * $p < 0.05$, **$P < 0.005$, ***$p < 0.0005$. Breg, regulatory B cell; dendritic cell; PBMC, peripheral blood mononuclear cell; WT, wild-type.

evaluating the effect of the cKO on the T cell population (S4I–S4O Fig and S12 Data) in naïve mice. The down-regulation of CD74 expression was specific to the DC population (S4H Fig, S12 Data, and gating strategy in S5A-S5K Fig). In contrast to the limited effect of CD74 deficiency in B cells on the tumor load, deficiency of CD74 in DCs remarkably reduced tumor growth. Mice deficient in CD74 in the CD11c population developed significantly smaller tumors compared to WT mice (Fig 5A–5C). These mice showed a reduced accumulation of IL-10[+] monocytes and macrophages and no difference in their IL-12 release (S6A-S6D Fig and S13 Data). In addition, DCs lacking CD74 in the TME displayed a decrease in their IL-10 levels (Figs 5D and S7A and S14 Data), demonstrating a direct role for CD74 in the regulation of the tol-DC phenotype, Bregs (Figs 5E and S7B and S14 Data) and Tregs (Figs 5F and S7C and S14 Data) and an increase of CD8[+] T cells in the TME (Figs 5G and S7D and S14 Data) characterized by a more cytotoxic and a less exhausted phenotype (Figs 5H, 5I, S7E, and S7F and S14 Data). No differences were detected in the frequency of CD4[+] T cells (S7G Fig and S14 Data) and in the frequency of CD8+ CD103+ or CD8+ CD62L+ cells (S7H and S7I Fig and S14 Data).

To further validate the effect of CD74 knockdown in DCs on B cells, in-vitro cocultures were established. Splenic DCs or B cells purified from the CD11c cKO mice and cultured with the E0771 cell line revealed a reduction of both tol-DCs (Fig 5J) and Bregs (Fig 5K), an effect that was not observed in the CD23 cKO mice. Thus, blocking of CD74 in CD11c[+] cells reduced the levels of tol-DCs, Bregs, and Tregs and their immunosuppressive functions in vivo and in vitro. Taken together, these results suggest that CD74 expressed in DCs, but not in B cells, is crucial for the regulation of immune cell suppression at the tumor site.

## CD74 expressed by DCs mediates the cross-talk with the Bregs

To further assess the CD74-mediated cross-talk between tol-DCs and Bregs, WT or CD74-deficient DCs and B cells were cultured together in vitro in the presence of E0771 cells. CD74 deficiency in both cell types reduced the expansion of tol-DCs (Fig 6A) and Bregs (Fig 6B) in a synergistic manner. However, while the lack of CD74 in the B cell population did not affect the Bregs or the tolerogenic DC, DCs lacking CD74 reduced not only the expansion of the tol-DCs (Fig 6A) but of the Bregs as well (Fig 6B), suggesting a role of CD74 in governing the Breg expansion. To determine whether DCs directly regulate Breg expansion, WT, and CD74[-/-] DCs were cultured with E0771 cells for 24 hours. Naïve B cells were then seeded with the previously activated DCs. As shown in Fig 6C, the lack of CD74 in DCs strongly reduced IL-10 release in B cells, further supporting the role of CD74 on DCs in mediating Breg expansion.

To understand whether B cells can affect the DC phenotype, B cells were incubated with the LN-2 anti-CD74 blocking antibody, or IgG control, and were activated with E0771 cells for 24 hours. Naïve DCs were then seeded with the previously activated B cells. CD74 inhibition did not regulate the expansion of tol-DC (Fig 6D).

To further confirm the immunosuppressive role of CD74 in DC activity, their function in T cell proliferation and suppression was analyzed. WT naïve splenic CD3[+] T cells were stained with the Cell Proliferation Dye (CPD), and then cocultured with WT or CD74 KO splenic DCs, previously activated with E0771 cells. Down-regulation of CPD expression in T cells correlates with the proportion of cells that undergo division. Induced CD8[+] T cell proliferation was detected in the cells incubated with CD74 KO DCs (Fig 6E). Moreover, lower levels of Tregs were observed in the presence of CD74KO DCs, further supporting an immunosuppressive role of CD74 expressed in DCs in the context of TNBC (Fig 6F). In addition, the cytotoxicity of CD8[+] T cells was assessed by analyzing interferon gamma (IFN-γ) protein levels. Elevated levels of IFN-γ were detected in T cells cocultured with CD74-deficient DCs,

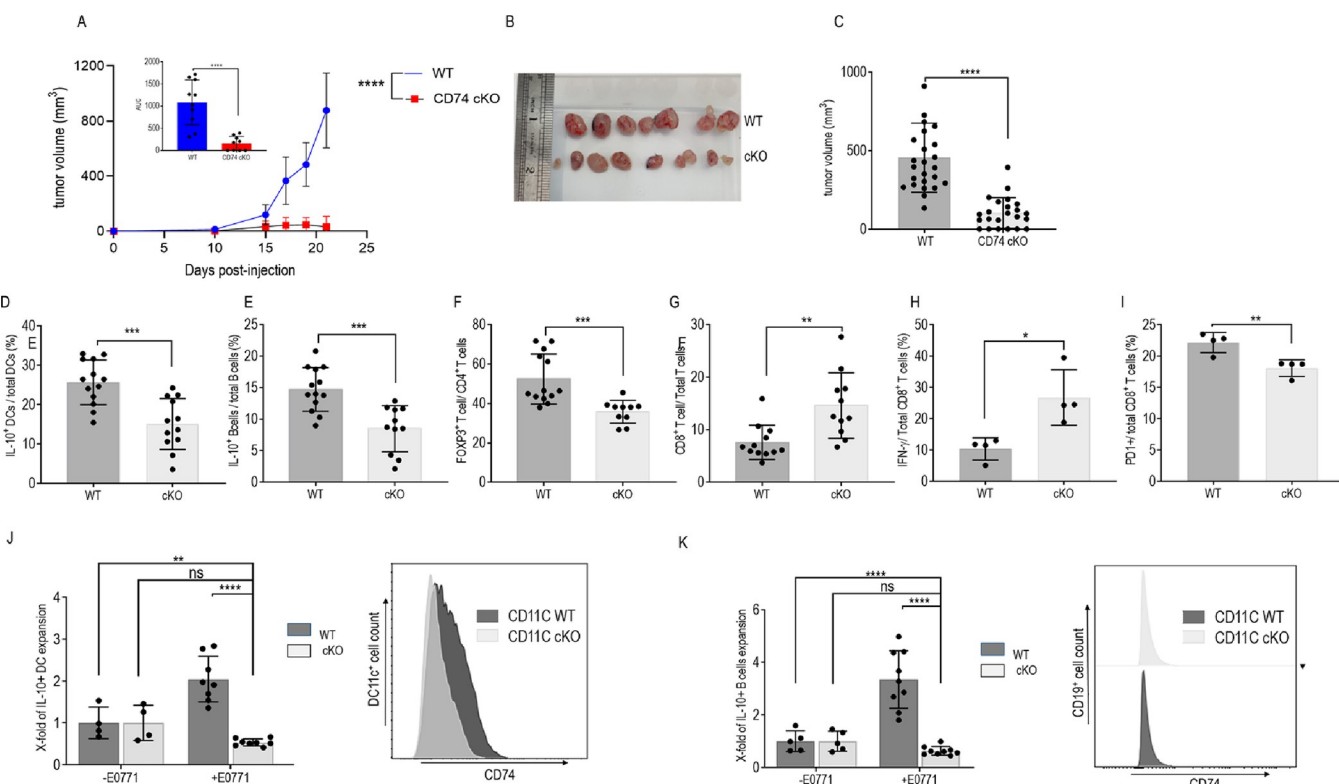

**Fig 5. CD74 deficiency in DCs reduces tumor proliferation by activation of the immune response. (A–I; S5 Data)** Female 6-week-old CD11c-Cre x CD74flox x CD74 flox mice were injected with 5 * 10⁵ E0771 cells into each of the fourth mammary pads. (**A**) Tumor size, recorded every 5 days. (**B**, **C**) After 21 days, tumor sizes were measured and recorded ($n = 44$); each dot represents a tumor. (**D**) IL-10⁺ DCs were analyzed as a percentage of total DCs (WT, $n = 13$; conditional cKO, $n = 12$). (**E**) Mean percentage and SD of IL-10⁺ B cells out of total B cells (WT, $n = 12$; cKO, $n = 11$). (**F**) Mean percentage and SD of Tregs out of total CD4⁺ cells (WT, $n = 13$; cKO, $n = 10$). (**G**) Mean percentage and SD of the tumor-infiltrating CD8⁺ T cells out of total CD3⁺ T cells (WT, $n = 13$; cKO, $n = 12$). (**H**) Mean percentage and SD of the IFN-γ releasing T cells out of total CD8⁺ T cells (WT, $n = 4$; cKO, $n = 4$). (**I**) Mean percentage and SD of PD1⁺ T cells out of total CD8⁺ T cells (WT, $n = 4$; CKO, $n = 4$) (**J, K; S5 Data**) Splenic B cells and DCs were cultured with E0771 cells at a 1:5 ratio. (**J**) Mean percentage and SD of IL10⁺ DCs out of total DCs (WT, $n = 8$; conditional CD74 KO, $n = 8$). (**K**) Mean percentage and SD of Bregs out of B cells (WT, $n = 8$; conditional CD74 ⁻/⁻, $n = 8$). The histogram represents the expression of CD74 in the DC and B cell populations. ns $p > 0.05$, * $p < 0.05$, ***$P < 0.0005$, ****$p < 0.00005$. Breg, regulatory B cell; DC, dendritic cell; WT, wild-type.

indicating stronger cytotoxic potential (Fig 6G and 6H). Furthermore, WT T cells were cultured in the presence of either WT or CD74 KO DCs, together with E0771 cells. To better understand the direct tumor-killing activity of T cells, perforin and granzyme B expression levels were analyzed. In parallel, the viability of E0771 in the coculture was assessed by Annexin-7AAD staining. T cells activated by CD74 KO DCs released higher levels of perforin (S8A and S8C Fig and S15 Data) and granzyme B (S8B and S8D Fig and S15 Data) as shown in Fig 6I and supported less viability of E0771 cells (Fig 6J) and greater apoptosis, compared to the T cells activated by WT DCs (S8E–S8G Fig and S15 Data). Thus, CD74 regulates the functionality of DCs in the TME.

## CD74 binds to the IL-1β and SP1 promotors, which, in turn, regulate tol-DCs and Breg expansion

We next investigated the mechanism of action of CD74 expressed on DC. TNBC cells were injected into the mice to form tumors. Starting from day 10, mice were intravenously treated for 5 consecutive days (days 10 to 14) with either PBS or DRQ. DCs were sorted from the TME, and purified RNA was then analyzed by RNA-seq. Inhibition of CD74 led to stronger

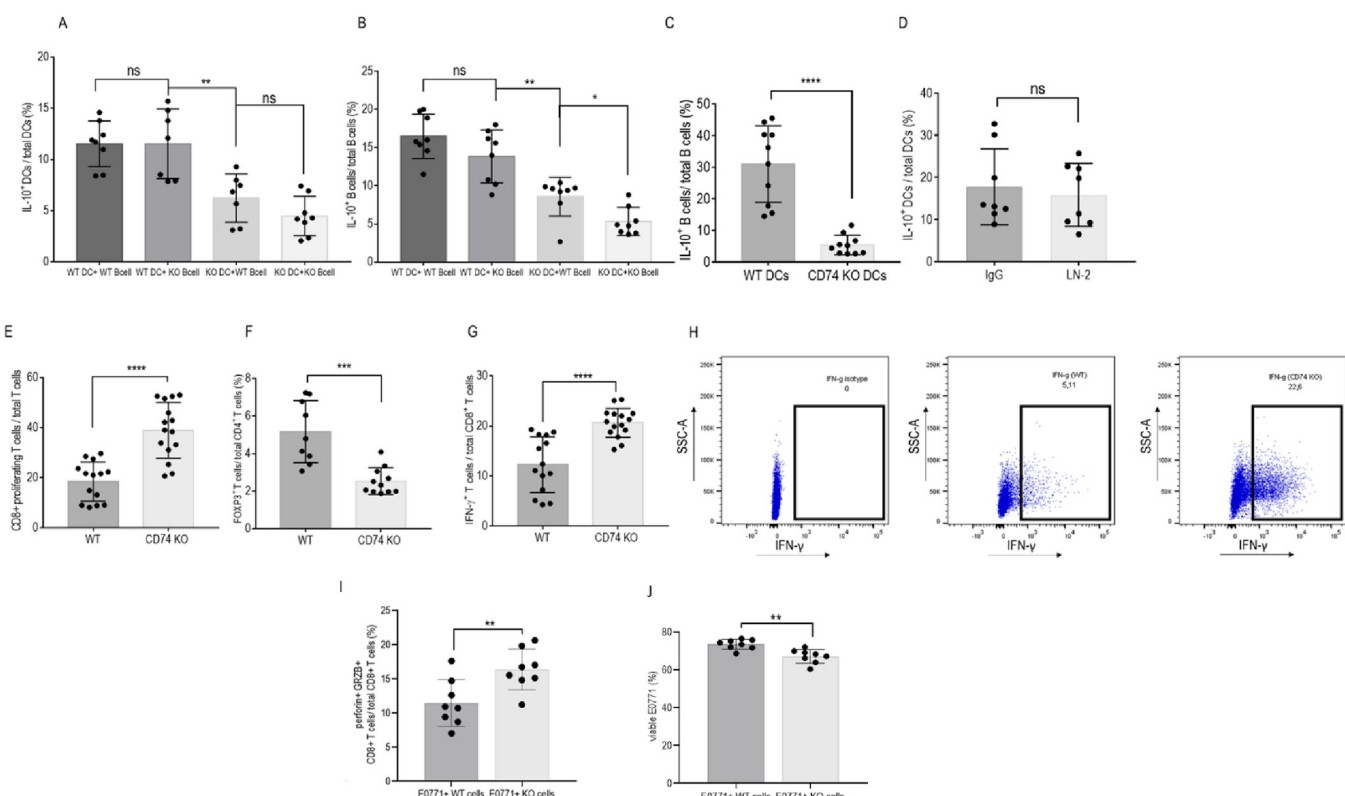

**Fig 6. CD74 mediates the cross-talk between DC and B cells, enhancing immunosuppression in the TME. (A, B; S6 Data)** B cells and DCs were isolated from spleens of naïve C57BL/6 and CD74$^{-/-}$ mice and cultured at a 1:1 ratio; E0771 cells were added at a 1:5 ratio. After 24 hours, B cells or DCs were collected and analyzed by flow cytometry. PIM activation was conducted before FACS staining. Dead cells were excluded from analysis by Zombie Live/Dead staining. (**A**) Mean percentage and SD of IL10$^+$ dendritic cells out of total dendritic cells, under the different culture conditions (WT DCs+ WT B cells, $n = 8$; CD75T4 KO DCs+ WT B cells, $n = 8$; WT DCs+ CD74 KO B cells, $n = 8$; CD74$^{-/-}$ DCs+ CD74 KO B cells, $n = 8$). (**B**) Mean percentage and SD of IL10$^+$ B cells out of total B cells, under the different culture conditions (WT DCs+ WT B cells, $n = 8$; CD74 KO DCs+ WT B cells, $n = 8$; WT DCs+ CD74 KO B cells, $n = 8$; CD74 KO DCs+ CD74 KO B cells, $n = 8$). (**C; S6 Data**) Splenic DCs were purified from CD11c-Cre x CD74flox x CD74 flox mice and activated with E0771 for 24 hours. Naïve splenic B cells were purified from C57BL/6 mice and activated with the WT or CD74$^{-/-}$ DC for 24 hours with fresh medium. Bar graphs depict the mean percentage and SD of IL10$^+$ B cells out of total B cells (B cells+WT DCs CM, $n = 10$; B cells+CD74 cKO DCs CM, $n = 10$). (**D; S6 Data**) Splenic B cells were purified from C57BL/6 mice, activated with the E0771, and incubated with an anti-CD74 blocking antibody (LN-2) or isotype control antibody for 24 hours. Naïve splenic DCs were purified from C57BL/6 mice and cultured together with the treated B cells with a fresh medium. Bar graph depicts the mean percentage and SD of IL10$^+$ DCs out of total DCs (DCs +B cells treated with Igg, $n = 8$; DCs+ B cells treated with LN-2, $n = 8$). (**E-H; S6 Data**) Splenic DCs were purified using magnetic beads from C57BL/6 and CD74$^{-/-}$ mice and cocultured with E0771 at a 1:3 ratio for 24 hours. Naïve Splenic CD3$^+$ T cells were purified from C57BL/6 mice and stained with the CPD to determine their proliferation. Following coculture with E0771, DCs were cultured with T cells at a 1:1 ratio in the presence of anti-CD3 antibody in the media for 72 hours. T cells were analyzed for CD8, FOXP3, and INF-γ. Dead cells were excluded from analysis by Zombie Live/Dead staining. (**E**) Mean percentage and SD of proliferating CD8$^+$ T cells under the different culture conditions (WT, $n = 14$; CD74 KO, $n = 15$). (**F**) Mean percentage and SD of FOXP3$^+$ CD4$^+$ T cells under the indicated culture conditions (WT, $n = 9$; CD74$^{-/-}$, $n = 11$). (**G**) Bar graphs depicting the percentage of IFN-γ$^+$ CD8$^+$ T cells (WT, $n = 14$; CD74 KO, $n = 15$). (**H**) Representative dot plots of IFN-γ releasing cytotoxic T cells after coculture with WT or CD74 KO DCs. (**I, J; S6 Data**) Naïve T cells were cultured in the presence of either WT or CD74 -/- DCs and E0771 for 48 hours. The ability of CD8+ cells to release perforin and GrzB was evaluated by FACS. (**I**) CD74 -/- DCs better educated the T cells to release perforin and GrzB compared to the WT DCs. (**J**) Furthermore, T cells previously educated by CD74 -/- DCs, reduced the viability of E0771, suggesting an increased killing of the E0771 cells.**$P < 0.005$, ***$P < 0.0005$, ****$p < 0.00005$. CPD, Cell Proliferation Dye; DC, dendritic cell; IFN-γ, interferon gamma; TME, tumor microenvironment; WT, wild-type.

DC activation, as seen by the up-regulation of key pathways in the antitumor immune response, such as the cross-talk between DCs and natural killer (NK) cells [41, 42] or the classical markers of macrophage activation, which are essential for the antitumor response [43]. Moreover, pathways involved in metabolic signaling important for cancer progression, such as the PPAR cascade, were down-regulated [44] (S9A Fig).

Furthermore, this analysis identified the upstream regulators responsible for gene expression changes observed in the DRQ-treated mice. The IL-10 receptor was down-regulated in DCs treated with the CD74 inhibitor (S9B Fig). Analysis of the downstream events showed

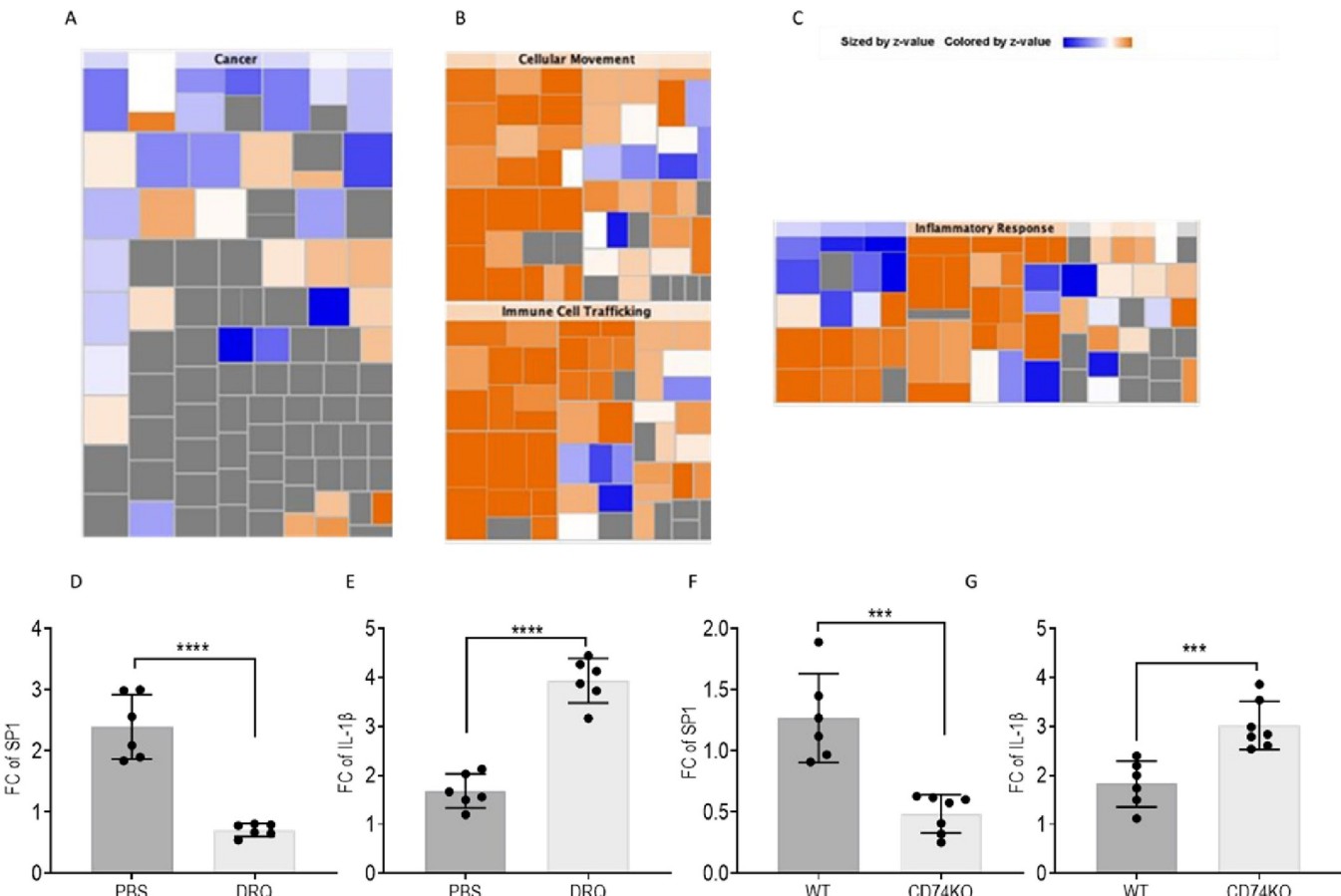

**Fig 7. CD74 deficiency in DCs induces pro-inflammatory pathways boosting the antitumor immune response.** Female 6-week-old C57BL/6 mice were injected with 5 * 10$^5$ E0771 cells into each of the fourth mammary pads. DRQ was injected intravenously on days 10, 11, 12, 13, and 14, after tumor onset. After 21 days, tumor sizes were measured and mice were killed. Tumors were processed into a single-cell suspension, and DCs were sorted from the TME of mice treated either with PBS or DRQ. Four replicates were analyzed for each group. **(A–C, S7 Data)** Heatmap of diseases, disorders, and biological functions of all differentially expressed genes from the RNA-seq data of DRQ vs. PBS (generated by IPA). Colors were defined by the z-scores, with orange representing up-regulation and blue down-regulation. Gene involved in cancer progression were down-regulated in the DRQ-treated samples, while genes related to immune trafficking, cellular movement, and inflammatory response were up-regulated. **(D, E; S7 Data)** mRNA levels of SP1 and IL1β were validated by qRT-PCR. Graph presents fold change of selected genes in the PBS-treated DCs versus DRQ-treated DCs (n = 6). **(F, G; S7 Data)** Fold change of SP1 and IL1β in WT and CD74 KO mice (n = 6). ***$p < 0.0005$ ****$p < 0.00005$. DC, dendritic cell; TME, tumor microenvironment; qRT-PCR, quantitative reverse transcription PCR; WT, wild-type.

elevation of a pro-inflammatory response, dictated mostly by the release of pro-inflammatory cytokines from DRQ-treated DCs (S9C Fig). In addition, blocking CD74 resulted in the down-regulation of cancer-related diseases (Fig 7A) and activation of the migration and interaction of the immune cells (Fig 7B and 7C). Among the genes differentially expressed in DRQ-treated DCs, SP1 was down-regulated, as confirmed by quantitative reverse transcription PCR (RT-qPCR) (Fig 7D), and IL-1β mRNA levels were up-regulated under these conditions (Fig 7E). An analogous effect was observed in DCs lacking CD74 (Fig 7F and 7G).

SP1 is associated with immunosuppression in several cancer types, including TNBC [45,46], and IL-1β is responsible for the activation of a pro-inflammatory pathway. Although IL-1β expression in the TME is related to cancer progression [47], DCs expressing IL-1β are more immunogenic, and reduce the expansion of immune-suppressive cells [48].

To verify that IL-1β controls IL-10 expression in DCs and the immunosuppressive ME, WT DCs were cultured in the presence of E0771 cells and incubated with either IL-1β or PBS. IL-

1β inhibited IL-10$^+$ DC expansion (Fig 8A and 8B). Moreover, to determine whether IL-1β-stimulated DCs regulate Breg expansion, B cells were cultured together with the DCs previously activated with E0771 and treated with either IL-1β or vehicle. As shown in Fig 8C and 8D, IL-1β treatment of DCs diminished Breg expansion, confirming that IL-1β treatment rendered DCs more immunogenic and consequently less capable of inducing Bregs. Thus, DCs have a direct effect on Breg expansion, a process that is attenuated by IL-1β release. Furthermore, to assess whether B cells are directly affected by IL-1β, B cells were cultured in the presence of E0771 and treated with either IL-1β or vehicle (Fig 8C). As shown in Fig 8D, IL-1β reduced Breg expansion. To determine whether the effect is DC-mediated, DCs were added to the B cell culture with E0771 cells (Fig 8E). The presence of DCs together with B cells strongly up-regulated IL-10 release, suggesting that DCs control Breg expansion in the presence of cancer cells and that IL-1β plays a key role in reducing IL-10 release (Fig 8F).

CD74-ICD is a regulator of transcription in health and disease [20,49]. To determine whether CD74 attenuates the expression of IL-1β in DCs, DCs derived from the TME were sorted, and the binding of CD74-ICD to the promotor region of IL-1β was analyzed by chromatin immunoprecipitation qPCR (ChIP-qPCR). A significant enrichment of binding of CD74-ICD to the promotor area of IL-1β in DCs was detected (Fig 8G), indicating that CD74-ICD regulates IL-1β transcription.

Next, we followed the role of SP1 in the immunosuppressive environment. To verify that SP1 contributes to IL-10 secretion by DCs, WT DCs were cultured in the presence of E0771 cells and incubated with either the SP1 blocker, mithramycin (MIT), or DMSO. MIT treatment abrogated IL-10$^+$ DC expansion, suggesting a direct correlation between SP1 and IL-10 release (Fig 9A).

To determine whether SP1 can directly induce Breg expansion via DCs, B cells were cultured together with DCs previously activated with E0771 and treated with either MIT or DMSO control (Fig 9B). MIT-treated DCs negatively affected Breg expansion, indicating that DCs control Breg expansion, a process that is augmented by SP1. To assess the role of SP1 on B cells, these cells were cultured in the presence of E0771 and treated with either MIT or DMSO. Blocking SP1 on B cells alone did not affect Breg expansion. However, the addition of DCs to this culture enhanced the percentage of Bregs and therefore boosted the effect of MIT on IL-10 release (Fig 9C). Finally, to determine whether CD74 regulates the transcription of SP1 in DCs, sorted DCs from the TME were activated for 1 hour with either PBS or MIF, and binding of CD74-ICD to the promotor region of SP1 was determined by ChIP-qPCR analysis (Fig 9D). MIF activation induced a significant enrichment of binding to the promotor regions of SP1 by CD74-ICD, indicating a direct effect on SP1 transcription through the MIF-CD74 axis (Fig 9E). Since we showed that IL-1β transcription is inhibited by CD74, we wished to determine whether SP1 acts as a transcription factor, in a MIF-CD74-dependent manner, to control IL-1β expression. SP1 bound the IL-1β promotor upon MIF activation, down-regulating its transcription (Fig 9F), but did not bind a noncoding region in the gene body (Fig 9H), confirming that SP1 specifically binds the IL-1β promotor. In addition, blocking SP1 up-regulated IL-1β expression (Fig 9G). These results suggest that CD74-ICD in DCs binds the SP1 promotor, which regulates IL-1β and IL-10 release, and, in turn, governs the expansion of the Breg population.

## Discussion

Expression of CD74 is significantly up-regulated in different cell types in various cancers [24–29] and is correlated with tumor progression. MIF is a pro-inflammatory cytokine that serves as the ligand of CD74 and consists of 2 homologues; MIF-1 (MIF) and MIF-2/DDT. MIF-1

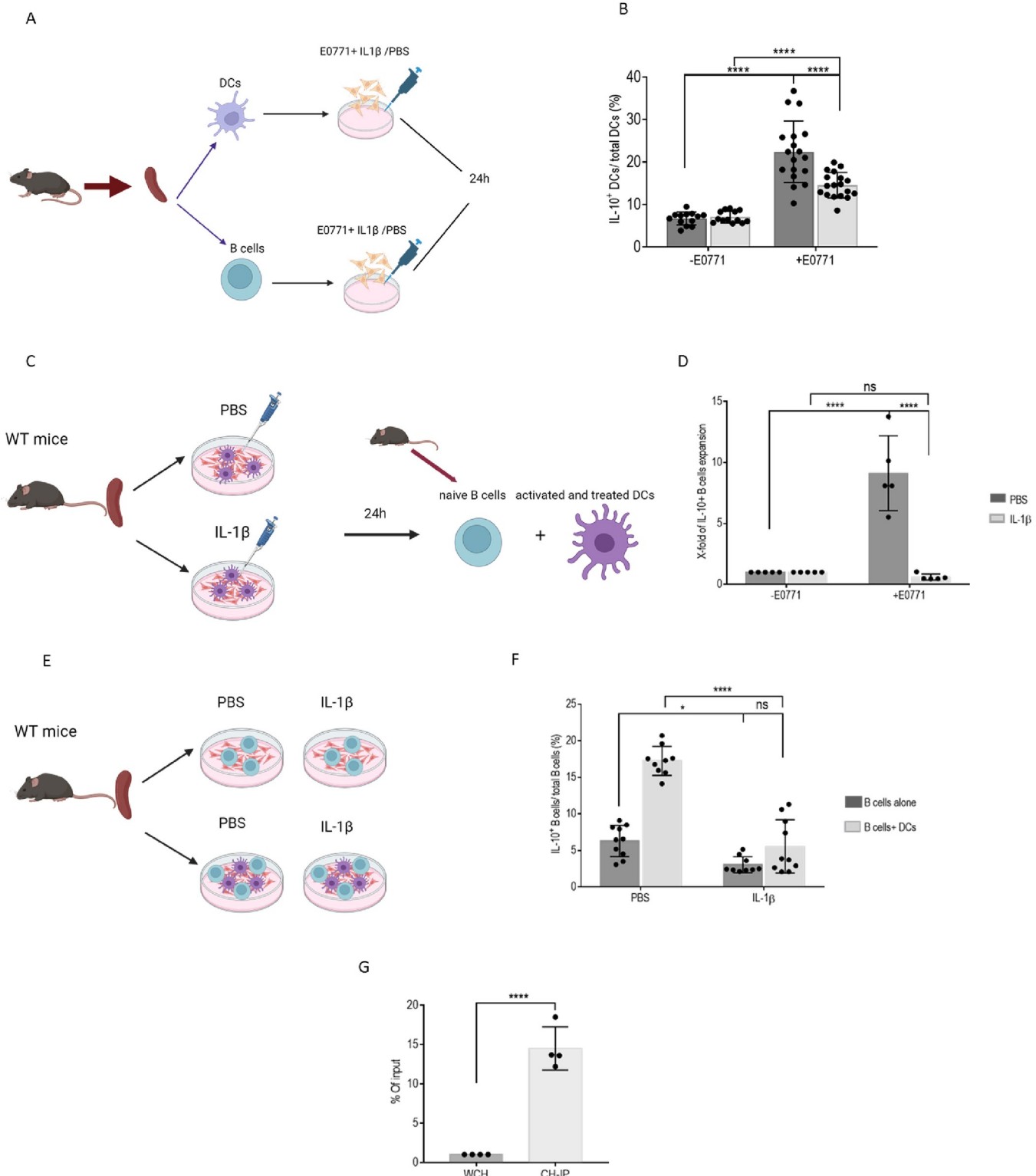

**Fig 8. CD74-ICD binds the IL-1β promotor in DC, promoting their tolerogenic phenotype. (A, B; S8 Data) (A)** Splenic DCs were isolated from vert-x mice and cultured in the presence of the E0771 at a 1:3 ratio. IL-1β agonist or vehicle was added to the cells for 48 hours. (**B**) Mean percentage and SD of IL10+ DCs out of total DCs (DCs+ vehicle, $n = 18$; DCs+ IL1β, $n = 20$). Each dot represents a mouse. **(C, D; S8 Data) (C)** Splenic DCs were purified from C57BL/6 mice and activated with the E0771 for 24 hours in the presence of PBS or IL-1β. Naïve splenic B cells were purified from C57BL/6 mice and added to the DCs for 24 hours after replacing the medium. (**D**) Bar graphs depict the fold change of IL10+ B cells out of total B cells, (B cells+ DCs treated with PBS, $n = 5$; B cells+ DCs

treated with IL-1β, *n* = 5). (**E, F**; **S8 Data**) (**E**) Splenic B cells were purified from C57BL/6 mice and activated with E0771 cells either alone or with purified DCs for 24 hours, in the presence of IL-1β or vehicle. (**F**; **S8 Data**) Bar graphs depict the mean percentage and SD of IL10$^+$ B cells out of total B cells (B cells treated with PBS, *n* = 10; B cells treated with IL-1β, *n* = 10; B cells + DCs treated with PBS, *n* = 10; B cells + DCs treated with IL-1β, *n* = 10). (**G**; **S8 Data**) Female 6-week-old C57BL/6 mice were injected with 5 * 10$^5$ E0771 cells into each of the fourth mammary pads. After 21 days, mice were killed. Tumors were processed into a single-cell suspension, and DCs were sorted from the TME. ChIP analysis was performed. The binding of CD74-ICD to the promotor area of the IL-1β gene was determined by qPCR. The graph represents the percentage of enrichment of the input (the amount of DNA pulled down by using the antibody of interest in the ChIP reaction, relative to the amount of starting material-input sample) (*n* = 4). ns *p* > 0.05, * *p* < 0.05, ****$p$ < 0.00005. CD74-ICD, CD74 intracellular domain; DC, dendritic cell; TME, tumor microenvironment.

exhibits higher affinity to CD74 compared to MIF-2, and, therefore, is a better candidate as a therapeutic target. MIF was previously shown to be necessary for the immunosuppressive ME in melanoma and glioblastoma [50,51]. It was found to be overexpressed in several types of tumors, including TNBC ([52,53] #2790). The MIF–CD74 axis has been shown to play pivotal roles, not only in initiating an oncogenic signaling pathway but also in provoking inflammatory responses, thereby promoting tumor growth and an immunosuppressive milieu [54–56]. Moreover, CD74 was shown to play a crucial role in dendritic cells and macrophages in the context of metastatic melanoma and glioma, reprogramming these cells to become tolerogenic and shift towards an M2 polarization [56,57]. In addition, CD74 function in DCs is associated with their motility, affecting their ability to migrate to sites of their activity [58]. However, the function of CD74 in the environment of the immune cells in the TNBC ME has not been described, and the specific function of DCs and Bregs in this context was not analyzed.

In the current study, we demonstrate that TNBC cells secrete MIF, which binds CD74 expressed on DCs and B cells, inducing a phenotypic switch from immunogenic to tolerogenic. Blocking CD74 leads to a reduced tumor load due to the elevated activity of the tumor-infiltrating immune cells. This antitumor phenotype is mainly caused by decreased IL-10 secretion, which results in a global decrease of the suppressive Bregs, Tregs, and tol-DCs in the TME.

Bregs and tol-DCs in the TNBC ME can also affect each other, enhancing IL-10 release via a positive feedback loop [59]. A feedback loop between tol-DCs and Bregs was previously demonstrated in several processes, such as T cell clonal anergy and Treg expansion, highlighting the complexity of the mediators involved in the generation and maintenance of tolerance. This cross-talk can be a double-edged sword, beneficial, for example, in the case of autoimmunity, but harmful in the case of cancer [60–62].

We show that DCs significantly govern Breg expansion via CD74-induced pathways. However, Bregs are not able to control DC differentiation, supporting the hypothesis that DCs are the major players among tumor-infiltrating immune cells. It is well established that tumor-infiltrating immune cells communicate among themselves with a synergistic effect [63].

Here, we show that DCs control B cells and induce their immunosuppressive phenotype by regulating the expression of several genes involved in the immune response, resulting in the upregulation of IL-10 expression. We suggest that activation of CD74 results in the binding of CD74-ICD, which serves as a transcription regulator [20,49], to the SP1 and the IL-1β promotors in DCs, and elevates their mRNA expression, suggesting that besides its role in B cells, CD74-ICD serves as a transcription regulator in DCs. Higher expression levels of SP1 [45], reduced expression of the pro-inflammatory cytokine IL-1β [64], and elevation of IL-10 secretion from the DC population are associated with immunosuppression in TNBC. Accordingly, these DCs are less immunogenic and enhance the expansion of immune-suppressive cells [48]. Although CD74 was shown to be associated to a pro-inflammatory response in several diseases, including pancreatic cancer [65–67], its role is context-dependent and depends on specific environmental cues. Indeed, it was demonstrated that CD74 regulates an anti-inflammatory pathway in melanoma promoting a tolerogenic phenotype of both DCs and macrophages; supports the regulatory T cell accumulation in several tumors; and hampers the

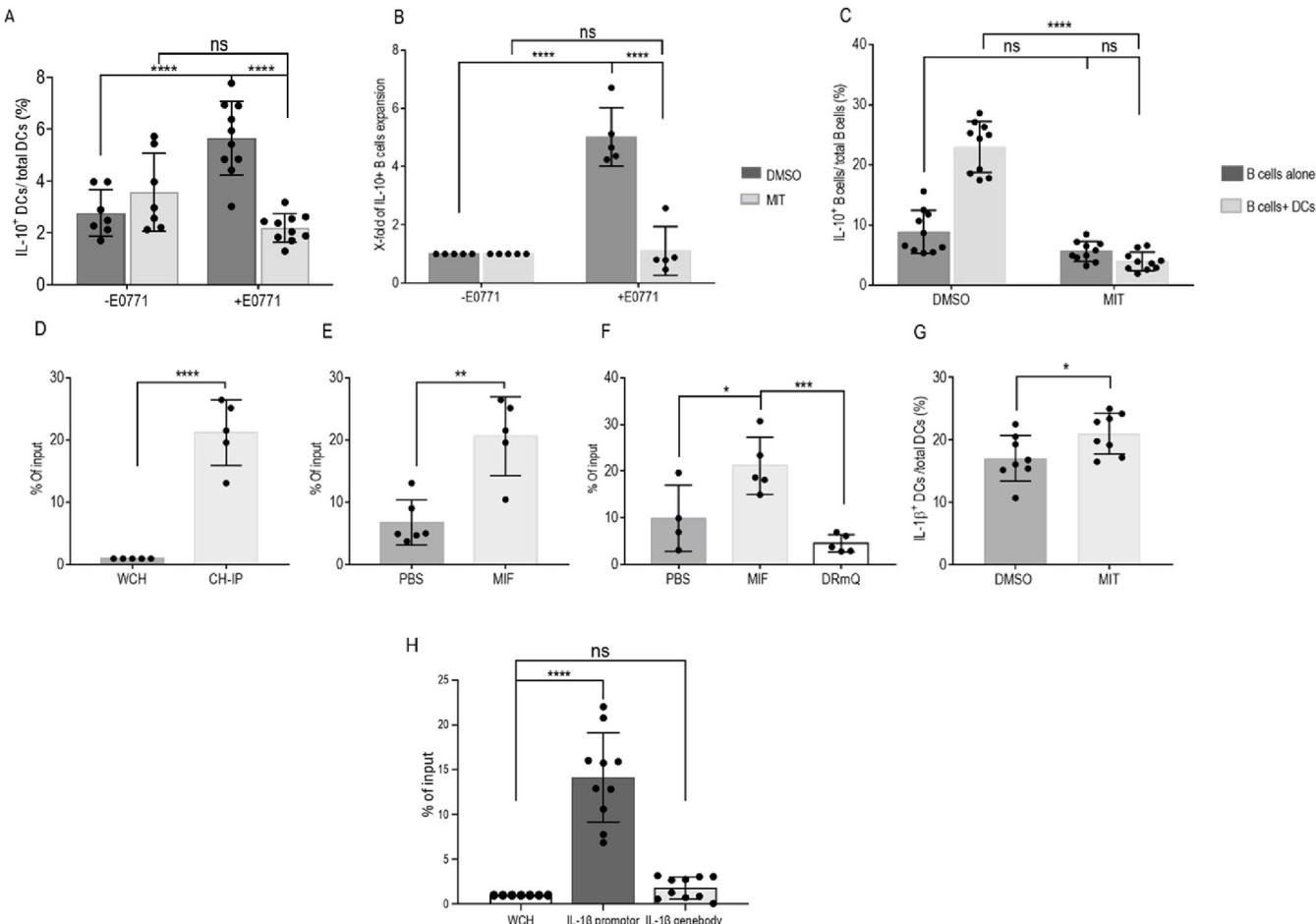

**Fig 9. SP1 binds the IL-1β promotor on DC via the MIF-CD74 axis, inducing their tolerogenic phenotype. (A–C; S9 Data)** Splenic dendritic cells were isolated from vert-x mice and cultured in presence of E0771 cells at a 1:3 ratio. SP1 blocker (MIT) or DMSO were added to the cells for 48 hours. (**A**) Fold change of the expansion of IL10$^+$ DCs out of total DCs (DCs+ DMSO alone $n = 7$, DCs+ MIT alone $n = 7$, DCs+ DMSO+ E0771, $n = 10$; DCs+ MIT+ E0771, $n = 10$). Each dot represents a mouse. (**B**) Splenic DCs were purified from C57BL/6 mice and activated with the E0771 for 24 hours in the presence of DMSO or MIT. Naïve splenic B cells were purified from C57BL/6 mice and added to the DCs for 24 hours after replacing the medium. Bar graphs depict the fold change of IL10$^+$ B cells out of total B cells (B cells + DCs treated with DMSO alone, $n = 5$; B cells + DCs treated with MIT alone, $n = 5$, B cells + DCs treated with DMSO+ E0771 $n = 5$, B cells + DCs treated with MIT+ E0771, $n = 5$). (**C**) Splenic B cells were purified from C57BL/6 mice and activated with either E0771 cells alone or with purified DCs for 24 hours in the presence of MIT or DMSO. Bar graphs depict the mean percentage and SD of IL10$^+$ B cells out of total B cells (B cells treated with DMSO, $n = 10$; B cells treated with MIT, $n = 10$, B cells+ DCs treated with DMSO, $n = 10$; B cells+ DCs treated with MIT, $n = 10$). (**D, E; S9 Data**) Female 6-week-old C57BL/6 mice were injected with 5 * 10$^5$ E0771 cells into each of the fourth mammary pads. After 21 days, tumor sizes were measured, and mice were killed. Tumors were processed into a single cell suspension, and DCs were sorted from the TME. Sorted DCs were activated for 1 hour with either mrMIF or vehicle, and a ChIP-qPCR for the SP1 promotor was performed. (**D**) The graph represents the percentage enrichment of the input (the amount of DNA pulled down by CD74 antibody in the ChIP reaction, relative to the amount of starting material-input sample) (WCH $n = 5$, SP1 promotor area $n = 5$). (**E**) Binding of CD74-ICD in DCs activated with mrMIF ($n = 5$). (**F; S9 Data**) Female 6-week-old C57BL/6 mice were injected with 5 * 10$^5$ E0771 cells into each of the fourth mammary pads. After 21 days, tumor sizes were measured and mice killed. Tumors were processed into single cell suspension, and DCs were sorted from the TME. Sorted DCs were activated for 1 hour with either mrMIF, DRQ, or vehicle, and a ChIP-qPCR for IL-1β promotor was performed. The graph represents the percentage of enrichment of the input (the amount of DNA pulled down by using SP1 antibody in the ChIP reaction, relative to the amount of starting material-input sample) in DCs activated with mrMIF, DRQ, or vehicle ($n = 14$). (**G; S9 Data**) The graph represents the percentage of SP1 binding to the promotor area of IL-1β compared to the genebody, a noncoding area of IL-1β (WCH, $n = 6$, IL-1β promotor area, $n = 10$, IL-1β genebody, $n = 10$). (**G; S9 Data**) DCs were purified from C57BL/6 mice and seeded together with E077 in the presence or absence of MIT. The graph represents the mean percentage and SD of IL-1β$^+$ DCs out of total DCs (DCs treated with DMSO, $n = 8$, DCs treated with MIT, $n = 8$). $p > 0.05$, $^*$ $p < 0.05$, $^{**}P < 0.005$, $^{***}P < 0.0005$, $^{****}p < 0.00005$. CD74-ICD, CD74 intracellular domain; ChIP-qPCR, chromatin immunoprecipitation qPCR; DC, dendritic cell; MIF, migration inhibitory factor; MIT, mithramycin; mrMIF, mouse recombinant MIF; TME, tumor microenvironment; WCH, whole cell extract.

M1 polarization of the microglia in brain metastasis from non-small cell lung cancer (NSCLC) [56,68,69].

Our results define the role of IL-1β specifically in the DC population within the tumor microenvironment, where it is known to regulate their inflammatory activity through IL-12 production. DCs lacking CD74 are able to promote an immunogenic response via IL-1β release, inhibiting the expansion of Bregs. In our model, the effect of DCs-mediated IL-1β secretion appears to be dominant compared to its direct effect on the B cells themselves, suggesting that IL-1β releasing DCs hamper the expansion of the Breg population. Studies on the role of IL-1β in regulatory B cells were performed mainly in autoimmune diseases, in which the environment and cytokines are completely different from those in the TME [70]. It is known that cytokines have different functions in different contexts and cells, and we therefore suggest that IL-1β might function differently in immune cells in the context of autoimmunity or cancer.

In conclusion, we show that blocking the CD74-induced pathway down-regulates SP1 expression in DCs, resulting in up-regulated IL-1β secretion, which strongly reduces Breg expansion, leading to the activation of the immune response. The increased IL-1β secretion from DCs is determined by both the direct effect of CD74 on its promotor, and the indirect effect of the MIF-CD74-SP1 axis.

These findings suggest that CD74 might serve as a novel therapeutic target in triple-negative breast cancer.

## Materials and methods

### Mice

C57BL/6, CD74$^{-/-}$, Vert-x, CD23-cre x CD74-flox, CD11c x CD74-flox mice were used in this study. Vert-x mice were provided by C. Mauri, University College London (UCL). All animals were used at 6 to 8 weeks of age. In the breast cancer model, only females were used, and the groups were age and sex-matched in each experiment. All animal procedures were approved by the Animal Research Committee at the Weizmann Institute of Science. To generate Cre-CD23 x flox-CD74 littermates, Cre-CD23 and flox-CD74 mice were crossed, and screened by PCR for CD74 and CD23 genotypes. To generate Cre-CD11c x flox-CD74 littermates, Cre-CD11c, and flox-CD74 mice were crossed and screened for CD74 and CD11c genotypes by PCR. IACUC 05690621–1.

### Breast cancer induction

E0771 cell-line cells were grown in a complete RPMI medium with 10% fetal bovine serum. For tumor models, $5 * 10^5$ cells in PBS were injected s.c. to each of the fourth mammary pads (total 2 mammary pads/mouse) of 6- to 8-week-old C57BL/Vert-x competent female mice.

### Tumor load measurements

Tumor size was assessed by external measurement of the length (L) and width (W) of the tumors in 2 dimensions using a Vernier caliper. Tumor volume (V, expressed in mm$^3$) was calculated using the following equation: $V = (L \times W^2/2$, when W is the shorter dimension measurement, and L is the longer).

### Preparation of tumor-infiltrating lymphocytes (TIL)

Tumor tissues were harvested 21 days following tumor implantation, cut into small pieces, and incubated in digestion buffer (1 mg/ml collagenase A, 0.15 mg/ml Hyaluronidase, 10% FBS,

1% P/S) for 45 minutes in a 37˚C incubator with gentle shaking. Tumor tissue was then passed through a 100 μm cell strainer and washed 3 times with PBS. Dissociated tumors were then suspended in 8 ml 44% Percoll solution and loaded onto 5 ml 67% Percoll cushions. Samples were centrifuged for 20 minutes at 1,000 RCF with no brake at room temperature. The middle fraction containing infiltrating mononuclear cells was collected and washed twice with PBS.

### B cell isolation from spleen and bone marrow

Murine spleens were dissected post-mortem and collected in PBS. Organs were processed through a 100-μm-cell strainer and treated with Red Blood Lysis buffer to lyse erythrocytes for 5 minutes. Next, cells were washed with PBS and processed through a 40-μm-cell strainer. Finally, B cells were purified by positive B cell selection with B220 magnetic beads.

### Immune cell isolation from spleen

Murine spleens were dissected post-mortem and collected in PBS. Organs were processed through a 100-μm-cell strainer and treated with Red Blood Cell Lysis buffer for 3 minutes. Cells were then washed with PBS and processed through a 40-μm-cell strainer.

### Regulatory B cell activation

For detection of IL-10 on B cells, B cells at $2.5 \times 10^6$ cells/ml in a complete ISCOVE medium were cultured for 5 hours with PMA (100 ng/ml), Ionomycin (1 μg/ml), Monensin (1 μg/ml), and LPS (10 μg/ml).

### Tolerogenic DC cell activation

For detection of IL-10 on DC cells, DC cells at $2.5 \times 10^6$ cells/ml in complete ISCOVE's medium were cultured for 5 hours with PMA (100 ng/ml), Ionomycin (1 μg/ml), Monensin (1 μg/ml), and LPS (10 μg/ml).

### Co-cultures

E0771 cancer cells were seeded in 12-well plates. The next day, B cells were purified from splenocytes by positive B cell selection with B220 magnetic beads. B cells were then either cultured alone, or co-cultured in 12-well plates in complete ISCOVE's medium with 10% FBS for 24 hours at a ratio of 1:5 B cells/E0771 cells. Similarly, DC were purified from splenocytes by positive Mojosort mouse Pan Dendritic cell isolation kit and added to the E0771 or cultured alone in 12-well plates with 10% FBS complete RPMI medium for 24 hours at a ratio of 1:3.

The total number of cells in each well was $2.5 * 10^6$ under all conditions. For the last 5 hours of culture, cells were activated with PMA, Ionomycin, Monensin, and LPS.

### Flow cytometry staining

FACS analysis was performed using FACS Canto. FACS data analysis was performed using FlowJo software. Antibodies are listed in S1 Table, below. Cells were stained using specific antibodies for surface markers as previously described, followed by fixation, and permeabilization using BD Cytofix/Cytoperm commercial kit or eBioscience transcription factor staining buffer set. Cells were then stained with intracellular antibodies.

## Tissue preparation for immunohistochemistry

Blocks of breast tissue from TNBC patients and healthy donors were cut at 5-μm thickness using a rotary microtome, and the sections were mounted onto Superfrost Plus glass microscope slides. The slides were dried at 37˚C overnight. Slides were stored at 4˚C until use.

## Opal multiplexed IHC staining

Immunohistochemistry was performed on deparaffinized and rehydrated 5-μm thick paraffin-embedded sections using xylene and a decreasing concentration of ethanol (100%, 96%, and 70%). Endogenous peroxidase activity was blocked with 3% $H_2O_2$ and 1% HCL in methanol for 30 minutes, followed by heat-induced antigen retrieval in Tris-EDTA (PH = 9). For non-specific binding, sections were blocked with 20% NHS (Vector Labs -S-2000) and 0.1% Triton. When a secondary biotinylated antibody was used, an additional step of biotin blocking (Vector Labs kit SP-2001) was performed. Sections were incubated with primary antibodies, diluted in 2% NHS and 0.1% Triton, overnight. Incubation with secondary HRP or biotin-conjugated antibodies was followed by fluorescently labeled OPAL or streptavidin (Sigma-Aldrich). CD11c, CD19, and CD74 staining were double amplified using a biotinylated secondary antibody, followed by ABC kit (Sigma-Aldrich) and OPAL reagent. Antibodies were removed by 10-minute microwave treatment in Tris-EDTA (pH = 9), and the protocol was repeated from the blocking step. Nuclei were stained with DAPI.

## Multiplex fluorescence imaging

Multispectral imaging was performed using PhenoImager at 20× magnification (Akoya Biosciences, Marlborough, MA) according to the manufacturer's instructions. The multispectral acquired images were loaded into InForm software for unmixing and background subtraction (inForm v.3.0; Akoya Biosciences). In the inForm software, an unstained slide (without DAPI and OPAL staining) was loaded and regions with high autofluorescence signals were marked for processing. After processing, image tiles were stitched using QuPath software with the "merge multiple TIFF fields" script.

## CD74 blocking with DRQ-2 in vivo

Blocking of CD74 in vivo was performed using DRQ and 20 mM TRIS buffer, pH8.5 in saline as a control.

Treatment with DRQ or PBS was started at day 10 after tumor cell administration and continued for 5 consecutive days. The inhibitor or control were injected into the tail vein (100 μg/ 100 μl per mouse).

## CD74 blocking with LN-2 antibody in vitro

B cells were treated with LN-2 blocking antibody or IgG isotype control (150 μg/ml) for 24 hours. The total number of cells in each well was $5 * 10^6$ under all conditions.

## MIF activation

Cultures of $5 \times 10^6$ cells were activated with 150 ng/ml of MIF activator in 1 ml medium in a 24-well plate for 24 hours.

### In vitro DC suppression assay

DCs were isolated from the spleen of WT and CD74$^{-/-}$ mice through the positive Mojosort mouse pan dendritic cell isolation kit. Isolated DCs were cocultured for 24 hours with E0771 cancer cells at a 1:3 ratio. Splenic CD3$^{+}$T cells were isolated using the CD3$^{+}$ mouse-positive selection kit. T cells were labeled with Carboxy Fluorescein Succinimidyl Ester (CFSE) and seeded at ratios of 1:1 with DCs, in the presence of anti-CD3 coupled beads for 72 hours. Cells were then collected, and T cells analyzed for proliferation by FACS.

### SP1 blocking in vitro

E0771 cancer cells were seeded in 12-well plates. The next day, DCs were purified from splenocytes and then co-cultured with the tumor cells, at an E0771/DC cell ratio of 1:3. Cells were cultured in complete RPMI medium + 10% FBS, in the presence of 20 μm of Mithramycin, or DMSO as a negative control for 48 hours. The total number of cells in each well was $2.5 * 10^{6}$ under all conditions.

### IL-1β activation in vitro

E0771 cancer cells were seeded in 12-well plates. The next day, DCs were purified from splenocytes and then co-cultured with the cancer cells, at an E0771/DC cell ratio of 1:3. Cells were cultured in complete RPMI medium + 10% FBS, in the presence of 20 nM of IL-1β recombinant antibody, or PBS as negative control, for 48 hours. Total number of cells in each well was $2.5 * 10^{6}$ under all conditions.

### RNA extraction for high-throughput experiments and RNA-sequencing

Tumor-infiltrating DCs were sorted from dissociated TME tissues. mRNA was extracted from these cells using the Dynabead mRNA purification kit, and Illumina libraries were constructed from total mRNA using the bulk adaptation of the MARS-Seq protocol [71] for Illumina Tru-Seq RNA Sample Preparation v2 (Cat. no.RS-122–2002, Illumina) according to the manufacturer's instructions. Indexed samples were sequenced in an Illumina NextSeq High output HiSEq 2500 machine in single-read mode. STAR (2.7.3a) TopHat (v2.0.10) was used to align the reads to the Mus_musculus (GRCm39) and human genomes (hg19). Reads were counted based on annotations downloaded from Ensembl (release 106) on hg19 RefSeq genes using HTSeq-count (version 0.11.2) (v0.6.1p1). Differentially expressed genes were identified using DESeq2 with the betaPrior, cooksCutoff, and independent filtering parameters set to false. Raw $P$ values were adjusted for multiple testing using the procedure of Benjamini and Hochberg. Differentially expressed genes were determined by a $p$-adj of <0.05, and absolute fold changes >1.5 and max raw counts >10.

### RNA extraction and cDNA synthesis for RT-qPCR

Total RNA was isolated from cells using the TRI Reagent RNA Isolation Reagent, according to the manufacturer's instructions. For cDNA synthesis, 500 ng or 1 μg mRNA was used with the qScript cDNA Synthesis Kit, according to the manufacturer's instructions.

### qRT-PCR

qRT-PCR was performed on the Lightcycler 480. The program used was: 10' at 95˚C, followed by 45 cycles of amplification (95˚C for 10", 60˚C for 10", 72˚C for 10") and then cooling to 4˚C. Primers appear in S2 Table.

### siRNA transfection

siRNA was introduced by electroporation using a Nepagene (Ichikawa, Chiba, Japan) Super Electroporator NEPA21 Type II, using 2 mm gap cuvettes, with 20 µg of siRNA at 225 mv, 5 msec in 100 µl of OptiMem medium. After the transfection, the cells were resuspended in RPMI 1% FCS medium and incubated for 24 hours.

### ChIP qPCR

ChIP-seq was performed as previously described [20]. For each sample, $5 \times 10^5$ tumor-infiltrating DC cells were sorted and activated with rmMIF or with vehicle for 1 hour, then cross-linked with disuccinimidyl glutarate (DSG) and fixed. Chromatin was immunoprecipitated with anti CD74 or anti-SP1 antibodies and ChIP-DNA was processed. The samples were analyzed by qPCR for SP1 or IL-1β promotor.

### Statistical analysis

Data analysis was performed using Graphpad Prism (Version 7.0 f, GraphPad Software, Inc., La Jolla, CA, USA). For most experiments, the mean is provided together with SEM or SD. To determine the significance of the differences, we used ordinary or 2-way ANOVA and Student's *t* test, either one- or two-tailed and 2 ways, depending on the experiment. Results were deemed significant with a *P* value of 0.05 or less.

None of the material in this manuscript has been published or is under consideration for publication elsewhere.

## Supporting information

**S1 Fig. Gating strategy for IL-10+ DCs and B cells and for IL-12+ DCs. (A–M)** PBMCs from the tumor site were activated with PIM and then analyzed by flow cytometry. Dead cells were excluded from analysis by Zombie Live/Dead staining. **(A)** DC cells were analyzed for CD11c expression after excluding LY6-C+, F4/80+, and CD19. B cells were analyzed for CD19 after excluding LY6-C+, F4/80+, and CD11c. **(B–E)** IL-10+ expression on DCs was measured by comparing the non-activated for either WT and CD74 -/- samples with the ones activated with PIM. **(F–I)** IL-12+ expression on DCs was measured by comparing the non-activated for either WT and CD74 -/- samples with the ones activated with PIM. **(J–M)** IL-10+ expression on B cells was measured by comparing the non-activated for either WT and CD74 -/- samples with the ones activated with PIM. The FCS files uploaded to FlowRepository (http://flowrepository.org/id/FR-FCM-Z8ES).
(TIF)

**S2 Fig. (S10 Data). CD74 regulates the accumulation of tolerogenic immune cells in the TME. (A–I)** 6 weeks old C57BL/6 and CD74-/- female mice were injected with 5 * 105 E0771 cells into each of the fourth mammary pads (total of 2 mammary pads per mouse). **(A)** DC cells were analyzed for CD45, CD11c, and IL-10 expression after excluding LY6-C+, F4/80 + and CD19+ cells. Graph shows the frequency of IL-10+ DCs in the tumor site (WT *n* = 14; CD74-/- *n* = 11). **(B)** Frequency of IL-10+ B cells out of total B cells (WT *n* = 8; CD74 -/- *n* = 7). **(C)** Frequency of IL-10+ macrophages after excluding monocytes and DCs (WT *n* = 5; CD74-/- *n* = 5). **(D)** Frequency of CD4+ T-cells (WT *n* = 9; CD74-/- *n* = 8). **(E)** Frequency of CD8+ T-cells (WT *n* = 9; CD74-/- *n* = 8). **(F)** Frequency of FOXP3+ T cells out of total CD4 + T cells (WT *n* = 9; CD74-/- *n* = 8). **(G)** Frequency of IFN-γ+ T cells out of total CD8+ T cells (WT *n* = 5; CD74-/- *n* = 4). **(H)** Frequency of PD1+ T cells out of total CD8+ T cells (WT *n* = 5; CD74-/- *n* = 4). **(I)** Frequency of CD62L+ T cells out of total CD8+ T cells (WT *n* = 5;

CD74-/- $n$ = 5). ns $p$ > 0.05, * $p$ < 0.05, ** $p$ < 0.005, **** $p$ < 0.00005.
(TIF)

**S3 Fig. (S11 Data). The CD74 blocker DRQ restores the immunogenicity of the TME. (A–E)** Six weeks old C57BL/6 female mice were injected with 5 * 105 E0771 cells into each of the fourth mammary pads (total of 2 mammary pads per mouse). On days 10, 11, 12, 13, 14, after tumor implantation, DRQ was intravenously injected. **(A)** DC cells were analyzed for CD45, CD11c, and IL-10 expression after excluding LY6-C+, F4/80+ and CD19+ cells. Graph shows the frequency of IL-10+ DCs in the tumor site (PBS $n$ = 8; DRQ $n$ = 8). **(B)** Frequency of IL-10 + B cells out of total B cells (PBS $n$ = 7; DRQ $n$ = 8). **(C)** Frequency of CD4+ T-cells (PBS $n$ = 4; DRQ $n$ = 5). **(D)** Frequency of FOXP3+ T cells out of total CD4+ T cells (PBS $n$ = 4; DRQ $n$ = 5). **(E)** Frequency of CD8+ T-cells (PBS $n$ = 4; DRQ $n$ = 5). ns $p$ > 0.05, * $p$ < 0.05, ** $p$ < 0.005.
(TIF)

**S4 Fig. (S12 Data). CD74 deficiency in DC specifically affects the dendritic cells population. (A-O)** Female 6-week- old CD11c-Cre x CD74flox x CD74 flox mice were killed, spleens were harvested, processed to a single cell suspension, and total PBMCs were isolated. **(A)** Mean and SD percentage of monocytes out of total live cells (WT $n$ = 5, cKO $n$ = 5). **(B)** Mean and SD percentage of macrophages out of total live cells (WT $n$ = 5, cKO $n$ = 5). **(C)** Mean and SD percentage of B cells out of total live cells (WT $n$ = 5, cKO $n$ = 5). **(D)** Mean and SD percentage of dendritic cells out of total live cells (WT $n$ = 5; CKO $n$ = 5). **(E–H)** Mean and SD percentage of CD74 expression on monocytes **(E)**, macrophages **(F)**, B cells **(G)**, and DCs **(H)** (WT $n$ = 6; cKO $n$ = 6). **(I)** Mean and SD percentage of CD4+ T cells out of total live cells (WT $n$ = 6; cKO $n$ = 6). **(J)** Mean and SD percentage of CD8+ T cells out of total live cells (WT $n$ = 6; cKO $n$ = 6). **(K)** Mean and SD percentage of CD62L+ T cells out of total CD4+ T cells (WT $n$ = 6; cKO $n$ = 6). **(L)** Mean and SD percentage of CD103+ T cells out of total CD8+ T cells (WT $n$ = 6; cKO $n$ = 6). **(M)** Mean and SD percentage of FOXP3+ T cells out of total CD4 + T cells (WT $n$ = 4; cKO $n$ = 4). **(N, 0)** Expression of CD74 in the CD4+ **(N)** and CD8+ **(O)** populations. ns $p$ > 0.05, ** $P$ < 0.005, *** $P$ < 0.0005.
(TIF)

**S5 Fig. CD26 and CD68 are not specific markers for DCs or macrophages. (A–K)** PBMCs from the spleen of naïve mice were analyzed by flow cytometry. **(A)** Dead cells were excluded from analysis by Zombie Live/Dead staining. **(B)** Macrophages and monocytes were gated for F4/80 and LY-6c, respectively. **(C)** The double negative population was analyzed for CD19 and CD11c to detect DC and B cells. **(D)** DCs obtained in panel C were analyzed for CD26. **(E, F)** The CD45+ population was gated for CD26 as a dendritic cell marker. **(G, H)** CD26+ DCs were analyzed for F4/80, LY-6c, and CD19. **(J)** The CD45+ population was gated for CD64 as a macrophage marker. **(K)** CD64+ macrophages were analyzed for CD19 and LY-6c expression. The FCS files uploaded to FlowRepository (http://flowrepository.org/id/FR-FCM-Z8ES).
(TIF)

**S6 Fig. (S13 Data). Conditional KO of CD74 in DCs impacts the IL-10 release from monocytes and macrophages. (A–D)** Female 6-week-old CD11c-Cre x CD74flox x CD74 flox mice were injected with 5 * 105 E0771 cells into each of the fourth mammary pads. After 21 days, mice were killed, and tumors were harvested, processed to a single cell suspension, and total PBMCs from the tumor site were isolated. Cells were then activated with PIM and analyzed by flow cytometry. **(A)** Mean and SD percentage of IL-10+ monocytes out of total monocytes (WT $n$ = 4, cKO $n$ = 4). **(B)** Mean and SD percentage of IL-12+ monocytes out of the total population (WT $n$ = 4, cKO $n$ = 4). **(C)** Mean and SD percentage of IL-10+ macrophages out of

total macrophages (WT $n = 4$, cKO $n = 4$). **(D)** Mean and SD percentage of IL-12+ macrophages out of total (WT $n = 4$, cKO $n = 4$). ns >0.05, * $P < 0.05$.
(TIF)

**S7 Fig. (S14 Data). CD74 conditional KO in dendritic cells reduces the frequency of tumor-infiltrating immunosuppressive cells. (A–I)** Female 6-week-old CD11c-Cre x CD74flox x CD74 flox mice were injected with 5 * 105 E0771 cells into each of the fourth mammary pads. After 21 days, mice were killed, and tumors were harvested, processed to a single-cell suspension, and total PBMCs from the tumor site were isolated. Cells were then activated with PIM and analyzed by flow cytometry. **(A)** Frequency of IL-10+ DCs in the tumor site (WT $n = 8$; CD74 cKO $n = 7$). **(B)** Frequency of IL-10+ B cells out of total B cells (WT $n = 7$; cKO $n = 7$). **(C)** Frequency of FOXP3+ T cells out of total CD4+ T cells (WT $n = 4$; CD74 cKO $n = 4$). **(D)** Frequency of CD 8+ T-cells (WT $n = 8$; CD74 cKO $n = 7$). **(E)** Frequency of IFN-γ+ T cells out of total CD8+ T cells (WT $n = 4$; CD74 cKO $n = 4$). **(F)** Frequency of PD1+ T cells out of total CD8+ T cells (WT $n = 4$; CD74 cKO $n = 4$). **(G)** Frequency of CD 4+ T-cells (WT $n = 7$; CD74 cKO $n = 8$). **(H)** Frequency of CD103+ T cells out of total CD8+ T cells (WT $n = 4$; CD74 cKO $n = 4$). **(I)** Frequency of CD62L+ T cells out of total CD8 + T cells (WT $n = 4$; CD74 cKO $n = 4$). ns $p > 0.05$, * $p < 0.05$, **** $p < 0.00005$.
(TIF)

**S8 Fig. (S15 Data). CD74 inhibition enhances the T cells killing activity. (A–G)** Naïve T cells were cultured in the presence of either WT or CD74 -/- DCs and E0771 for 48H. The ability of CD8+ cells to release perforin and GrzB was evaluated by FACS. **(A–D)** CD74 -/- DCs better educated the T cells to release perforin compared to the WT DCs (A, C), and GrzB (B, D). **(E–G)** Furthermore, T cells previously educated by CD74 -/- DCs, induced an increased killing of the E0771 cells, with major amount of both early and late apoptotic E0771 (F, G) * $p < 0.05$,, *** $p < 0.0005$, **** $p < 0.00005$. The FCS files uploaded to FlowRepository (http://flowrepository.org/id/FR-FCM-Z8ES).
(TIF)

**S9 Fig. CD74 down-regulation in DC induces pro-inflammatory pathways. (A–C)** RNA-seq analysis. Female 6 weeks old C57BL/6 mice were injected with 5 * 105 E0771 cells into each of the fourth mammary pads (total of 2 mammary pads per mouse). After 10 days, DRQ was injected intravenously for 4 consecutive days (days 10–13). After 21 days, tumor sizes were measured and mice killed. Tumors were processed into single-cell suspension and DCs were sorted from the tumor microenvironment of mice treated either with PBS or DRQ. Four replicates were used from each group. **(A)** Visualization of the Ingenuity Pathway Analysis (IPA) where the relevant pathways are shown ordered by significance ($p$ value), calculated in IPA by right-tailed Fischer's exact $t$ test. The pro-inflammatory pathways show a positive z-score indicating that pathway activity is increased in DRQ versus PBS-treated mice. **(B)** IPA Upstream Regulator Analysis was used to predict the upstream regulators responsible for the gene expression changes observed. IL-10 receptor is shown to be down-regulated in DCs treated with DRQ. **(C)** Image depicts the gene interactions, where genes shown in red are up-regulated, while genes shown in green are suppressed. Color intensity is relative to gene expression. Genes related to immunogenic response of DCs are increased in DRQ versus PBS treated mice.
(TIF)

**S1 Table. Antibodies list used for FACS analysis.**
(DOCX)

**S2 Table. qPCR primers list.**
(DOCX)

**S1 Data. Excel spreadsheet containing, in separate tabs, the numerical data underlying Fig 1.**
(XLSX)

**S2 Data. Excel spreadsheet containing, in separate tabs, the numerical data underlying Fig 2.**
(XLSX)

**S3 Data. Excel spreadsheet containing, in separate tabs, the numerical data underlying Fig 3.**
(XLSX)

**S4 Data. Excel spreadsheet containing, in separate tabs, the numerical data underlying Fig 4.**
(XLSX)

**S5 Data. Excel spreadsheet containing, in separate tabs, the numerical data underlying Fig 5.**
(XLSX)

**S6 Data. Excel spreadsheet containing, in separate tabs, the numerical data underlying Fig 6.**
(XLSX)

**S7 Data. Excel spreadsheet containing, in separate tabs, the numerical data underlying Fig 7.**
(XLSX)

**S8 Data. Excel spreadsheet containing, in separate tabs, the numerical data underlying Fig 8.**
(XLSX)

**S9 Data. Excel spreadsheet containing, in separate tabs, the numerical data underlying Fig 9.**
(XLSX)

**S10 Data. Excel spreadsheet containing, in separate tabs, the numerical data underlying S2 Fig.**
(XLSX)

**S11 Data. Excel spreadsheet containing, in separate tabs, the numerical data underlying S3 Fig**
(XLSX)

**S12 Data. Excel spreadsheet containing, in separate tabs, the numerical data underlying S4 Fig.**
(XLSX)

**S13 Data. Excel spreadsheet containing, in separate tabs, the numerical data underlying S6 Fig.**
(XLSX)

**S14 Data. Excel spreadsheet containing, in separate tabs, the numerical data underlying** S7 Fig**.**
(XLSX)

**S15 Data. Excel spreadsheet containing, in separate tabs, the numerical data underlying** S8 Fig**.**
(XLSX)

## Acknowledgments

The authors wish to thank members of the Shachar lab for fruitful discussion and support. Furthermore, the authors wish to thank Tali Shalit for her help. IS is the incumbent of the Dr. Morton and Ann Kleiman Professorial Chair.

## Author Contributions

**Conceptualization:** Richard Bucala, Idit Shachar.

**Data curation:** Bianca Pellegrino, Keren David, Stav Rabani, Bar Lampert, Thuy Tran, Edward Doherty, Marta Piecychna.

**Formal analysis:** Bianca Pellegrino, Keren David, Stav Rabani, Edward Doherty, Marta Piecychna, Shirly Becker-Herman, Idit Shachar.

**Funding acquisition:** Idit Shachar.

**Investigation:** Bianca Pellegrino, Keren David, Stav Rabani, Thuy Tran, Arthur A. Vandenbark, Shirly Becker-Herman, Idit Shachar.

**Methodology:** Roberto Meza-Romero, Lin Leng, Richard Bucala.

**Resources:** Roberto Meza-Romero, Lin Leng, Arthur A. Vandenbark, Richard Bucala.

**Writing – original draft:** Bianca Pellegrino, Idit Shachar.

**Writing – review & editing:** Arthur A. Vandenbark, Richard Bucala, Shirly Becker-Herman.

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
