## [Editor Report · Decision Letter 0]

18 Jan 2024

Dear Dr Shachar, 

Thank you for submitting your manuscript entitled "CD74 Regulates the Tumor Immunosuppressive Microenvironment in Triple-Negative Breast Cancer" for consideration as a Research Article by PLOS Biology. Please also accept my apologies for the delay, mainly due to the holiday period.

Your manuscript has now been evaluated by the PLOS Biology editorial staff as well as by an academic editor with relevant expertise and I am writing to let you know that we would like to send your submission out for external peer review.

Once your full submission is complete, your paper will undergo a series of checks in preparation for peer review. After your manuscript has passed the checks it will be sent out for review. To provide the metadata for your submission, please Login to Editorial Manager (https://www.editorialmanager.com/pbiology) within two working days, i.e. by Jan 22 2024 11:59PM.

Kind regards,

Ines

--

Ines Alvarez-Garcia, PhD

Senior Editor

PLOS Biology

---

## [Decision Letter · Decision Letter 1]

9 Apr 2024

Dear Dr Shachar,

Thank you for your patience while your manuscript entitled "CD74 Regulates the Tumor Immunosuppressive Microenvironment in Triple-Negative Breast Cancer" was peer-reviewed at PLOS Biology. Please also accept again my sincere apologies for the delay in providing you with our decision. The manuscript has now been evaluated by the PLOS Biology editors, an Academic Editor with relevant expertise, and by three independent reviewers. 

The reviews are attached below. As you will see, the reviewers are mostly positive and find the conclusions interesting and novel, however they also raise several concerns that would need to be addressed. Reviewer 1 thinks you should provide more human data, and to expand on the findings that loss of CD74 resulted in enhanced T-cell mediated cytotoxicity – suggests checking the relationship between CD74 expression and the amount of tolerogenic DCs and B cells in IHC/IF of human tumours, and including a T-cell killing assay to show how the loss of CD74 results in enhanced CD8+T cell activity. Reviewer 2 proposes several experiments to confirm the findings and to improve the figures and methods, along with the clarification of several points. Reviewer 3 thinks that some of the claims should be toned down, as they are not supported by the experiments, and also mentions that there are some missing references, and that further discussion of several points in the text is needed.

In light of the reviews, we would like to invite you to revise the work to thoroughly address the reviewers' reports. Given the extent of revision needed, we cannot make a decision about publication until we have seen the revised manuscript and your response to the reviewers' comments. Your revised manuscript is likely to be sent for further evaluation by all or a subset of the reviewers.

**IMPORTANT - SUBMITTING YOUR REVISION**

3. Resubmission Checklist

a) *PLOS Data Policy*

b) *Published Peer Review*

d) *Blurb*

Please also provide a blurb which (if accepted) will be included in our weekly and monthly Electronic Table of Contents, sent out to readers of PLOS Biology, and may be used to promote your article in social media. The blurb should be about 30-40 words long and is subject to editorial changes. It should, without exaggeration, entice people to read your manuscript. It should not be redundant with the title and should not contain acronyms or abbreviations. For examples, view our author guidelines: https://journals.plos.org/plosbiology/s/revising-your-manuscript#loc-blurb

Sincerely,

Ines

--

Ines Alvarez-Garcia, PhD

Senior Editor

PLOS Biology

Reviewers' comments

Rev. 1: Christina Cho – note that this reviewer has signed her review

Overall, the manuscript was an easy read--the findings were presented clearly and in a logical order. The authors did a good job of demonstrating a novel role for CD74--as a regulator of DC and B cell expansion. Although the idea of CD74 as a potential therapeutic target for TNBC is not a new one, the mechanism--CD74 promoting the formation of an immune-suppressed TME via increased expansion of tolergenic DCs--is new and exciting. A few minor suggestions to improve the paper: (1) Provide more human data and (2) expand on the findings that loss of CD74 resulted in enhanced T-cell mediated cytotoxicity. Perhaps the authors could have done some IHC/IF of human tumors demonstrating a relationship between CD74 expression and the amount of tolerogenic DCs and B cells. Also, in addition to the levels of IFNg, the authors could have included a T-cell killing assay to demonstrate how the loss of CD74 results in enhanced CD8+T cell activity. One last minor comment: Figure 6A is lacking a key to indicate what the bars represent, unless the key in figure 6B is also to be referenced for figure 6A.

Rev. 2:

General Comments:

Pellegrino et.al. presented a very interesting and original research paper describing the role CD74 expressed in dendritic cells on regulating the immune suppression in the tumoral microenvironment.

Still, I believe there are some experiments that need additional controls to be ready for publishing.

Major Comments:

-On Fig.1: It is not clear (also in Fig. 1 legend) which kind of data is the embedded subplot in 1A. Is it redundant with 1C? in this case leave 1C only.

- On page 5 it is claimed: "...., we next wished to determine whether the reduced tumor load detected in the CD74 deficient mice results from its function as a MIF receptor, or whether it is due to the role of CD74 in antigen presentation, controlling T cell numbers".

First, It is not clear if authors want to determine if the effect of CD74 full KO mice over tumor growth is a) a general effect of having less B and T cells "Since deficiency of CD74 results in a reduced number of mature B cells and CD4+ T cells [37]" as they describe, or b) antigen presentation is acutely involved in the anti-tumoral effect under the full CD74 KO. Authors should clarify this.

Second, to address if tumor growth depends on CD74 functioning as a MIF receptor and not as a regulator of antigen presentation, authors grow tumors in the presence of CD74 ligand inhibitor DRQ vs. control (veh). Authors show that DRQ treatment also inhibits tumor growth, as in the full KO mice, claiming that "These results suggest that MIF binding to CD74 positively regulates the tumor-suppressive ME in TNBC".

DRQ was previously used in the context of the nervous system as an EAE pre-clinical treatment. To make a claim respecting the role of CD74 as a MIF receptor and not as regulator of antigen presentation, authors should include bibliography respect the specificity of DRQ as a MIF binding inhibitor (and not over antigen presentation) or show experimentally that DRQ do not show any effect on antigen presentation, using an in vitro assay of antigen presentation with DCs.

- On Fig.2 authors show results regarding the dependency on CD74 of tol DC and reg B expansions in the presence of EO771 tumor cells. Authors should calculate the fold values of the -EO771 columns dividing each experimental result by the mean of each experiment to show data dispersion, as I guess they did for the +EO771 columns.

-On page 5, the authors claim" Thus, the MIF secreted from the malignant cells plays a crucial role in the regulation of tol-DC and Breg expansion". This claim comes from the observation that when co-culturing MIF deficient vs. control tumor cells, tol-DC and Breg expansion is dependant on tumoral cell MIF.

To claim "secreted", authors should show that the culture of B or DC cells with tumoral cell supernatant under these conditions (MIF depleted by siRNA vs. control), also reproduce the same dependencies on MIF regarding tol-DC and Breg expansions.

- In the ChIP experiments described in Fig.7D & 8 D-F, it will be very important to know if the SP1 signal is specific for the SP1 binding site of the promoter region of IL-1β. For this, a region of DNA that you do not expect to be enriched (e.g. a control region of the IL-1β gene where SP1 do not be expected to bind) and thus do not be expected to be amplified by qPCR, should be included in the procedure (negative control). A ratio of the signal found in the SP1 binding region over the one measured on the negative control region can be used as the SP1 binding measurement between the different experimental conditions. All statements regarding CD74 pathway regulating the transcriptional function of SP1 over IL1B depends on this ChIP control to be performed.

- In general, for all the figures in which the results of in vitro protocols are displayed: the described in vitro experiments have complex schemes and sometimes is not clear from the written descriptions the exact protocol used. It will help a lot to the reader to have in every figure that uses an in vitro assay a graphical scheme describing the different steps in each experiment. Some of these protocols are described in the methods section but are general and did not define the specifics of each experiment.

-In general, for all the plots were all control treatment measurements represented by dots are in the same Y level, equal to the mean (e.g.: 6D): why is so? Can you explain this? I think the correct way to proceed is the next one: Take the mean of all control measurements (mean control), if you divide every control measurement by the mean control, that will give you the data of relativized control measurements, which should have a dispersion. Same with the treatment measurements: divide every treatment measurement by the mean control and that will give you the data of relativized treatment measurements and now you can calculate the treatment mean. Also you can use this relativized data to calculate for statistical significance.

Minor Comments:

-On page 4 there is a typo, change pregulated to regulated.

- On page 5 it is claimed: "..., but the cytotoxic activity of CD8+T cells was induced in the TME of CD74 KO mice". It is not clear which Figure depicts this, please clarify.

Rev. 3:

MIF is an inflammatory cytokine and atypical chemokine that signals through CD74 and through the chemokine receptors CXCR2 and 4 to promote inflammatory and tumorigenic processes. Here, Pellegrino and colleagues have studied the specific role of the MIF/CD74 axis in triple negative breast cancer (TNBC) and in the tumor microenvironment. They demonstrate that CD74 expressed on CD11c+ cells (i.e. dendritic cells), regulates tumor growth through the control of the cross-talk between the tumor-infiltrating tolerant (tol)-DCs and regulatory B cells (Bregs). The study is comprehensive, well-controlled and the manuscript well written and of general interest for biologists and biomedical researchers. The findings suggest the MIF/CD74 axis as a novel therapeutic target in TNBC.

Main comments

- Upon MIF binding, CD74 forms a cell surface complex with CD44, which is essential for the MIF-induced signaling cascade. The signaling pathway involves Syk tyrosine kinase and PI3K/Akt activation, which leads to CD74 intramembrane cleavage and the release of the CD74 intracellular domain (CD74-ICD).

- Regarding a statement in the introduction and - generally - conceptually, it would be interesting for the reader to know whether the MIF-triggered CD74-ICD mechanism has actually been shown to involve CD44. In other words, is ICD generation blocked in Cd44-deficient cells? If this hasn't been shown, I would suggest to down-tone the phrase.

- Referencing needs to be improved and updated; some key papers from the MIF and CD74 field are missing. For example, a relatively recent paper reporting on CD74 in DCs is missing in the reference list. This paper should be included and accordingly discussed.

- Figure 1L ff.: based on the pharmacological blocking experiment with DRQ, authors suggest that "MIF binding to CD74 positively regulates the tumor-suppressive ME in TNBC". Can they please discuss (or explain in the rebuttal letter), as to how DRQ might also block MIF-2 responses, and how/why it doesn't interfere with the MHC II chaperone activity of CD74.

- Figure 2: the co-culture experiment in Figure 2 would highly benefit from an experimental flow chart ("cartoon") outlining the co-culture set up. I would suggest to include this scheme as new Figure 2A.

- Figure 2 ff.: the MIF siRNA approach in Figure 2F is an elegant approach to show that tumor cell secreted MIF is involved. Still as MIF knockdown may more generally alter tumor cell behavior, consider using (or at least discussing) the usual approaches to interfere with secreted MIF, i.e. using a neutralizing MIF Ab or MIF SMDs such as 4-IPP. Plus: were MIF levels in the sups actually controlled, e.g. by ELISA?

- Figure 6: I understand that normalization was necessary in these experiments. Still, consider representing the error bars for the PBS controls.

- Discussion: Cd74-deficiency leads to decreased numbers of Bregs, Tregs, and IL-10+ macrophages in the TME, overall suggesting that CD74 (on DCs) promotes an anti-inflammatory profile in the TME. Authors may want to discuss how this compares to the otherwise and generally known role of CD74 in promoting inflammation. In other words, it would be good to even better discuss the mechanisms that specifically lead to this kind of inverse phenotype in the specific setting of the TME.

Minor points

1. Page 4, typo: CD74 expression is pregulated in various....

2. Consider citing Verjans et al, BMC Cancer 2009; which shows specifically an involvement of MIF in BC.

3. Figure 8: MIF or MIT?

---

## [Decision Letter · Decision Letter 2]

30 Aug 2024

Dear Dr Shachar,

Thank you for your patience while we considered your revised manuscript entitled "CD74 Regulates the Tumor Immunosuppressive Microenvironment in Triple-Negative Breast Cancer" for publication as a Research Article at PLOS Biology. This revised version of your manuscript has been evaluated by the PLOS Biology editors, the Academic Editor and two of the original reviewers.

Based on the reviews, we are likely to accept this manuscript for publication, provided you satisfactorily address the policy-related requests stated below.

In addition, we would like you to consider a suggestion to improve the title:

"CD74 promotes the formation of an immunosuppressive tumor microenvironment in triple-negative breast cancer in mice by inducing the expansion of tolerogenic dendritic cells and regulatory B cells”

We expect to receive your revised manuscript within two weeks. 

*Published Peer Review History*

*Press*

Sincerely,

Ines

--

Ines Alvarez-Garcia, PhD

Senior Editor

PLOS Biology

ETHICS STATEMENT:

Thank you for providing the ethics statement. Please also include an approval number.

Fig. 1B-D; Fig. 2A, C-I, K-O; Fig. 3B, C, E, F, H-J; Fig. 4A-F; Fig. 5A, C-K; Fig. 6A-J; Fig. 7D-G; Fig. 8B, D, F, G; Fig. 9A-H; Fig. S1A-M; Fig. S2A-I; Fig. S3A-E; Fig. S4A-O; Fig. S5A-K; Fig. S6A-D and Fig. S7A-I

**For figures containing FACS data, we ask that you provide FCS files and a picture showing the successive plots and gates that were applied to the FCS files to generate the figures. If the files are too big, please submit them to the Flow Repository (http://flowrepository.org/), or similar, and make sure the data is made publicly available.

CODE POLICY

Reviewers' comments

Rev. 1: Christina Cho

The authors sufficiently addressed the comments I had. The additional figures help support their primary conclusions and I am convinced that CD74 has a role in the expansion of tolergenic DCs and the generation of an immune-suppressed microenvironment.

Rev. 2:

General Comments:

Pellegrino et.al. presented a very interesting and original research paper describing the role CD74 expressed in dendritic cells on regulating the immune suppression in the tumoral microenvironment.

The authors properly addressed major and minor concerns that were raised, some of them were answered via new experiments (ChIP experimental concerns) and the majority via clarifications in the presentation of the data figures/manuscript text.

From my standpoint, the manuscript is ready to be accepted.

---

## [Editor Report · Decision Letter 3]

18 Oct 2024

Dear Dr Shachar,

Thank you for the submission of your revised Research Article entitled "CD74 promotes the formation of an immunosuppressive tumor microenvironment in triple-negative breast cancer in mice by inducing the expansion of tolerogenic dendritic cells and regulatory B cells" for publication in PLOS Biology. On behalf of my colleagues and the Academic Editor, Albana Gattelli, I am delighted to let you know that we can in principle accept your manuscript for publication, provided you address any remaining formatting and reporting issues. These will be detailed in an email you should receive within 2-3 business days from our colleagues in the journal operations team; no action is required from you until then. Please note that we will not be able to formally accept your manuscript and schedule it for publication until you have completed any requested changes.

PRESS

Sincerely, 

Ines

--

Ines Alvarez-Garcia, PhD

Senior Editor

PLOS Biology
